# Hydrostatic pressure drives sprouting angiogenesis via adherens junction remodelling and YAP signalling
Dunja Alexandra Al-Nuaimi [1], Dominic Rütsche [2], Asra Abukar[1], Paul Hiebert[3,6], Dominik Zanetti[3], Nikola Cesarovic[4,5], Volkmar Falk[4,5], Sabine Werner [3], Edoardo Mazza [1,2] ✉ & Costanza Giampietro [1,2] ✉

Endothelial cell physiology is governed by its unique microenvironment at the interface between blood and tissue. A major contributor to the endothelial biophysical environment is blood hydrostatic pressure, which in mechanical terms applies isotropic compressive stress on the cells. While other mechanical factors, such as shear stress and circumferential stretch, have been extensively studied, little is known about the role of hydrostatic pressure in the regulation of endothelial cell behavior. Here we show that hydrostatic pressure triggers partial and transient endothelial-to-mesenchymal transition in endothelial monolayers of different vascular beds. Values mimicking microvascular pressure environments promote proliferative and migratory behavior and impair barrier properties that are characteristic of a mesenchymal transition, resulting in increased sprouting angiogenesis in 3D organotypic model systems ex vivo and in vitro. Mechanistically, this response is linked to differential cadherin expression at the adherens junctions, and to an increased YAP expression, nuclear localization, and transcriptional activity. Inhibition of YAP transcriptional activity prevents pressure-induced sprouting angiogenesis. Together, this work establishes hydrostatic pressure as a key modulator of endothelial homeostasis and as a crucial component of the endothelial mechanical niche.

The formation of new vessels from existing ones – angiogenesis – enables fundamental physiological and pathological processes, such as embryonic development, wound healing, tumor growth, atherosclerosis, and other cardiovascular diseases[1–3]. Biochemical cues like growth factors and cytokines have long been recognized as critical regulators of angiogenesis[4,5]. However, emerging evidence highlights the significance of mechanical forces in this process[6–9]. Mechanical stimuli exerted by the local extracellular matrix (rigidity and topography), shear stress, as well as stretch and fluid pressure, are known regulators of cellular processes involved in angiogenesis[5,10]. For example, transmural shear stress results in a pressure gradient from vessel lumen to local extracellular space, which triggers angiogenic sprouting via weakening of cell-cell adhesion[11,12]. Still, in addition to exerting shear stress in transmural flow, plasma fluid is simultaneously isotropically pressurized (hydrostatic pressure) in the microvasculature. Perhaps owing to the difficulty of untangling pressure and pressure-derived flows, only a few studies have explored the effect of hydrostatic pressure alone as an angiogenic stimulus. In general, the effect of pressure on endothelial cells is arguably the least researched contributor to the endothelial microenvironment[1].

In vivo, in vascular beds most associated with angiogenesis (i.e., capillaries), cells are subjected to static pressures of approximately $10-30$ mmHg, rather than pulsatile pressures (Fig.1a)[13–15]. Of note, it has previously been reported that among mammals, blood pressure values are similar, irrespective of animal size[16–19]. Despite not typically being discussed as a modulator of the angiogenic program, several studies document that hydrostatic pressure plays a role in its fundamental steps (Fig. 1b). Most prominently, hydrostatic pressure ranging from 2.9 to 120 mmHg has long been known to stimulate proliferation of endothelial cells[2,20–25].

[1]ETH Zürich, DMAVT, Experimental Continuum Mechanics, Zürich, 8092, Switzerland. [2]Empa, Swiss Federal Laboratories for Materials Science and Technology, Experimental Continuum Mechanics, Dübendorf, 8600, Switzerland. [3]Department of Biology, ETH Zürich, Institute of Molecular Health Sciences, 8093 Zürich, Switzerland. [4]Department of Cardiothoracic and Vascular Surgery, German Heart Center Berlin, 13353 Berlin, Germany. [5]Department of Health Sciences and Technology, ETH Zürich, 8093 Zürich, Switzerland. [6]Present address: Centre for Biomedicine, Hull York Medical School, The University of Hull, Hull, HU6 7RX, UK. ✉e-mail: mazza@imes.mavt.ethz.ch; costanza.giampietro@empa.ch

Furthermore, arterial levels of pressure (50−150 mmHg) have been reported to induce dynamic, time-dependent cytoskeletal reorganization[2,6] and lead to disrupted adherens junctions[2,26,27]. Despite its relevance, hydrostatic pressure is usually neglected in in vitro studies, while atmospheric pressure, which is the physiological equivalent to venous pressure conditions, is typically applied. Whether physiological capillary levels of pressure (mean pressure of 20 mmHg) contribute to angiogenesis remained unexplored in previous studies[1], although tubulogenic activity is enhanced under 20 mmHg[28] and 40−50 mmHg[29,30].

In this study we identified static capillary level (20 mmHg) but not venule (0 mmHg), or arteriole (55 mmHg) hydrostatic pressure as a subtle modulator of the angiogenic phenotype in vitro and ex vivo. Using a custom-made bioreactor, we applied and compared the effects of controlled levels of hydrostatic pressure to 2D endothelial monolayers from different vascular beds and 3D organotypic vessel models and ex vivo models (Fig. 1c). Our results demonstrate that hydrostatic pressure stimulation at levels characteristic for capillaries, triggers sprouting angiogenesis through induction of a transient and partial endothelial-to-mesenchymal transition

(EndMT) phenotype characterized by enhanced proliferation and migration. On the molecular level we show that pressurized cells undergo a transient increase in N-Cadherin expression, a structural weakening of the adherens junctions, and an increased Yes-Associated Protein 1 (YAP) expression, nuclear localization, and signaling. We also demonstrate that the inhibition of YAP transcriptional activity is sufficient to prevent hydrostatic pressure-induced sprouting angiogenesis.

## Results

### Capillary pressure values tune endothelial sprouting ex vivo

To test the effect of microvascular static pressure on sprouting angiogenesis, we performed an ex vivo 3D aortic ring assay, which is widely used to evaluate pro- and anti-angiogenic stimuli[31]. Since blood pressure is comparable in mice and humans[16–19,29], thoracic aortic rings isolated from wild-type C57BL/6 mice were embedded in a 3D collagen I matrix and maintained under static pressure. Samples were pressurized under 0 mmHg (control), 20 mmHg (mean capillary pressure) and 55 mmHg (peripheral arteriole pressure), respectively, for 4 days (Fig. 2a). Quantification of the

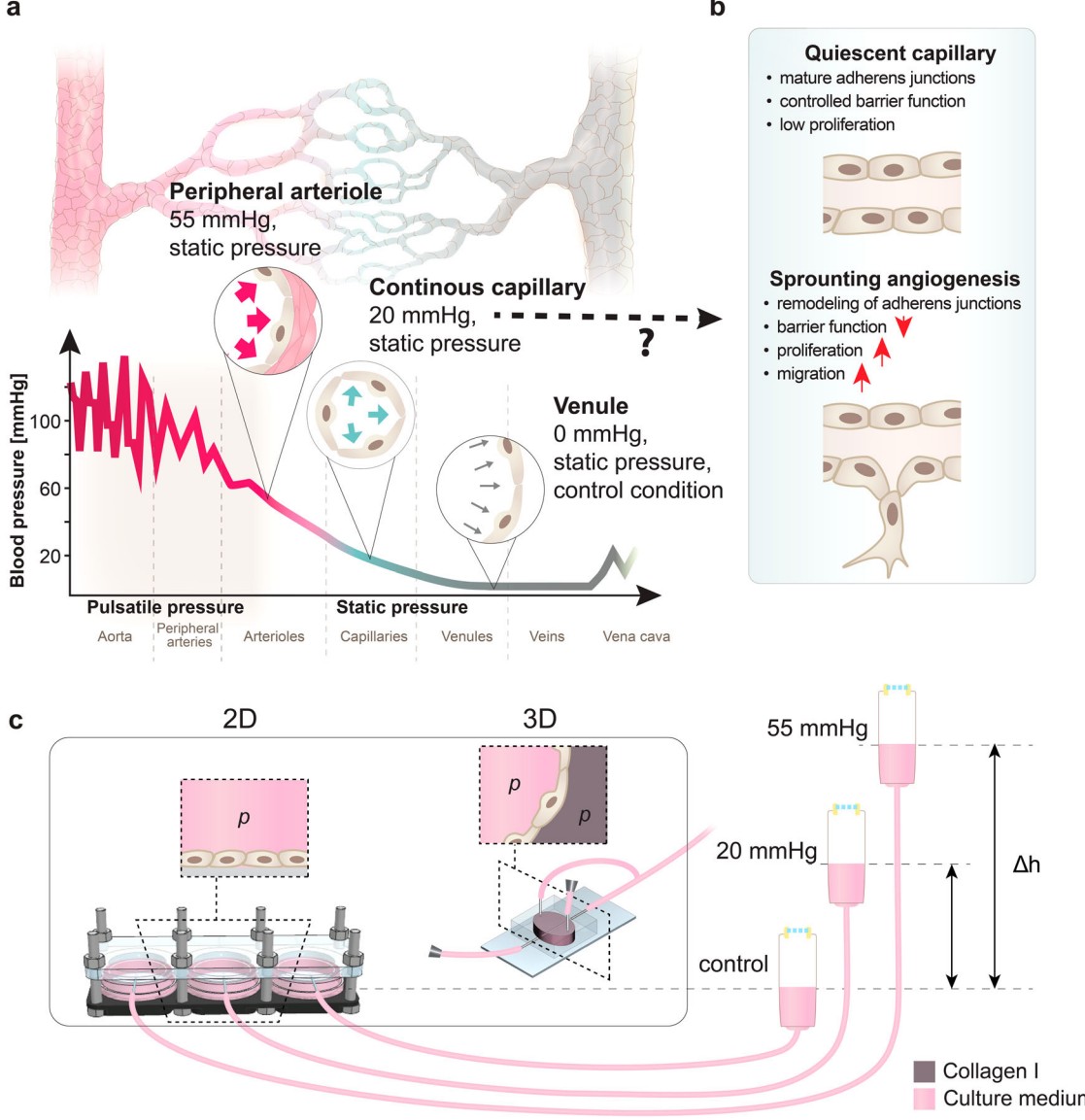

**Fig. 1 | Microvascular blood pressures applied to 2D and 3D organotypic endothelial cell cultures. a** Diagram of a vascular tree and corresponding physiological blood pressures where peripheral arterioles, capillaries and venules are subjected to static pressures. **b** Molecular and functional changes associated with

sprouting angiogenic vessels. **c** Schematic of 2D and 3D bioreactors used in this work to employ static hydrostatic pressure by means of culture media column. p: hydrostatic pressure.

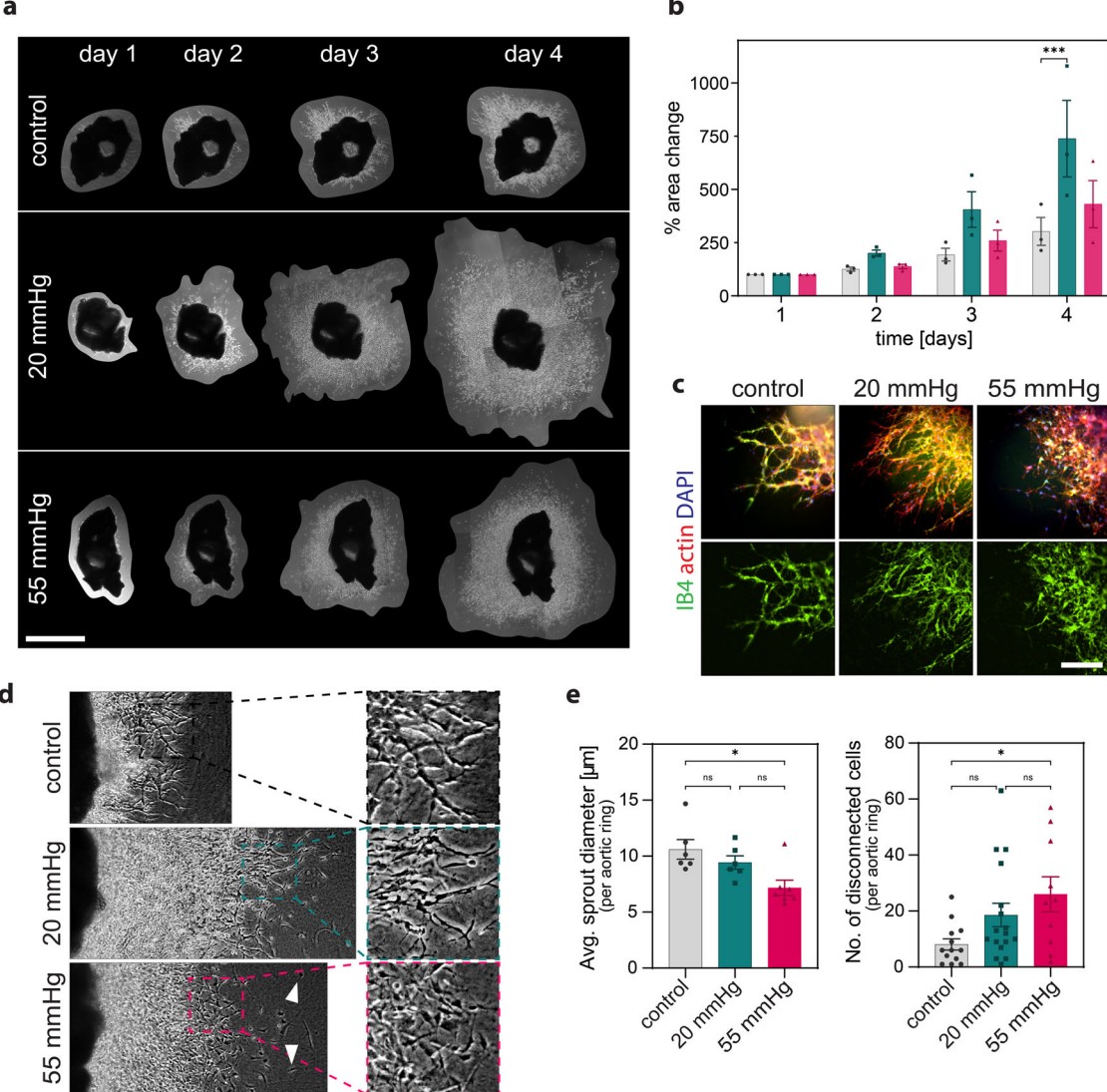

**Fig. 2 | Characterization of endothelial sprouting under hydrostatic pressure.** **a** Light microscope images of mouse aortic rings over time. **b** Quantification of sprout area growth over 4 days. $N = 3$ independent mouse experiments, $n = 3$ aortic rings per experiment. **c** Widefield z-projection images of aortic rings. Isolectin B4 (IB4, green) stains endothelial sprouts, actin (red), 4',6-diamidin-2-phenylindol (DAPI, blue). **d** Light microscope images of sprouts and disconnected cells (white arrows) after 4 days of pressure application. **e** Quantification of sprout diameter and of invasive cells disconnected from sprouts. $N \geq 6$ aortic rings per group. Data information: Graphs show mean ± s.e.m., ns = non-significant; *$p < 0.05$, ***$p < 0.001$ (**b**) 22 two-way ANOVA with Tukey's post-hoc test, (**e**) Kruskal-Wallis test. Scale bars: (**a**) 1000 μm, (**c**) 23 110 μm, (**d**) 100 μm.

endothelial sprout area, detected by immunostaining for the endothelial marker isolectin B4 (IB4) (Fig. 2c), revealed a differential growth across conditions starting from day 3. A substantial increase in area coverage by sprouts from aortic rings was observed under 20 mmHg (Fig. 2a, b). Quantification of the sprout diameter in the leading-edge regions on day 4 (Fig. 2d, e), revealed that sprouts formed under 55 mmHg pressure were significantly thinner compared to the static control. Pressure of 55 mmHg notably enhanced the number of cells disconnected from sprouts when compared to the control group (Fig. 2d, e), thus making sprouting less effective.

Collectively, these findings indicate that hydrostatic pressure at levels seen in capillaries (20 mmHg), but not arterioles (55 mmHg), promotes efficient sprouting in the ex vivo aortic ring assay. Moreover, hydrostatic pressure on the upper end of the mammalian physiological range[16] resulted in defective sprout formation, suggestive of abnormal vascularization.

Given our finding that varying levels of pressure can modulate the growth patterns and structural characteristics of newly developed sprouts,

we postulated that pressure affects cellular functions involved in sprouting angiogenesis.

**Hydrostatic pressure disrupts cellular homeostasis and initiates dynamic cellular responses**

Irrespective of the specific vascular bed, endothelial cells in a homeostatic in vivo environment maintain a quiescent state characterized by limited proliferation and migration, along with a tightly regulated barrier function[32]. Maintenance of this state is dynamically sustained as extrinsic and intrinsic signals are integrated continuously. Endothelial cells partially transition from their quiescent state to an activated state during sprouting angiogenesis, wherein they exhibit heightened proliferation and directional migration for sprout elongation and leader cell extension, respectively[13]. Considering the remarkably enhanced sprouting of aortic ring endothelial cells under pressure, we postulated that pressure alteration may affect cellular quiescence and drive endothelial cells towards an activated pro-angiogenic state. To test this hypothesis, we transitioned to a custom 2D in vitro model that

could be pressurized straightforwardly via a medium column pressure overhead (Fig. 1c). As the limited literature available regarding the effect of hydrostatic pressure on endothelial cells is mainly explored using human umbilical vein endothelial cells (HUVECs)[22,27,30,33,34], for reference and comparison purposes we initially used the same cell model.

To assess proliferation, we performed two complementary assays. We exposed matured HUVEC monolayers to pressure stimulation for 24 h, and we stained them for the nuclear cell-cycle marker Ki-67[35]. Compared to the unpressurized control, under 20 mmHg pressure there was a minor, but non-significant increase in Ki-67-positive cells that became statistically significant under 55 mmHg pressure, where the ratio of cells undergoing proliferation was approximately doubled (Fig. 3a, b). Notably, these results are in line with previous work in various pressure regimes[2,20–25]. To corroborate these results, we also quantified DNA synthesis in live cells. Fluorescent detection of EdU incorporation upon 24 h of pressurization further supported our observations, as we detected an increase in the percentage of EdU-positive cells under both pressure stimulations (Fig. 3a, b).

To determine the effect of hydrostatic pressure on directional collective cell migration, we performed a scratch wounding assay (wound area; Fig. 3c). Both levels of hydrostatic pressure slowed down the wound closure compared to the no pressure control (Fig. 3c, d), and 55 mmHg hydrostatic pressure resulted in a more disorganized and inefficient wound closure compared to the other conditions. To better understand these observations, we evaluated the trajectories of cells at the wound edge, and we observed a reduction of the x-displacement (towards wound closure) and an increase in the y-displacement (parallel to the wound edge) for the two pressure conditions compared to the control (Fig. 3e). Cells under pressure migrated in a significantly less directional manner (Fig. 3f), which corresponds to a higher directionality index.

Mature and quiescent endothelial monolayers exhibit reduced cell motility as a function of local cell density, known as jamming transition[36], regulated by mechanical and biochemical signal integration[37]. Conversely, the loss of quiescence is linked to an increase in the motility of individual cells, resembling an unjamming transition[38]. We thus tracked the trajectories of single endothelial cells within mature monolayers stimulated with hydrostatic pressure for 24 h. Interestingly, we observed a striking increase in the individual cell motility and cell displacement within the monolayers under pressure compared to the static control (Fig. 3g, h), which was pressure magnitude dependent. Increased proliferation and motility within the endothelial monolayer relies on the cells' ability to transiently remodel cell-cell adhesions, and vice versa, these adhesions hold a chief role in the regulation of endothelial functions such as barrier maintenance[39–41]. We thus evaluated barrier properties using a vascular permeability imaging assay that measures the leakage of streptavidin-488 from the junctions[42]. Endothelial monolayers of control samples stained for VE-Cadherin (Fig. 4a), the major adhesive protein of adherens junction complexes[43], showed few paracellular sites of leakage as previously reported[42,44], while both pressure-stimulated monolayers showed an increase in paracellular leakage (Fig. 4a, b). These results altogether suggest that hydrostatic pressure stimulation affects cell proliferation, directional migration, motility, and barrier properties of endothelial monolayers.

## Hydrostatic pressure stimulation modulates structural plasticity of the junctions and increases junctional N-Cadherin

Endothelial monolayers control cell proliferation, migration, and barrier function through the dynamic organization and remodeling of their cell-cell junctions[45], which are well known sensors and transducers of mechanical forces[46]. To investigate whether hydrostatic pressure affects junctional stability, we analysed the effect of pressure on the organization of VE-Cadherin based on immunostained images acquired at higher magnification[39,47–49]. As shown by immunofluorescence staining in Fig. 4c, hydrostatic pressure-stimulated cells showed a massive reorganization of adhesions, which acquired a jagged morphology[48,50]. By contrast, VE-Cadherin localization was straight in control cells, indicating the presence of stable junctions[39,47,48]. This change in junctional morphology was previously described in

HUVECs stimulated with similar values of hydrostatic pressure[27,34]. These results were further strengthened through the measurement of junction perimeter and area using the Feret's diameter, which was significantly increased in the samples stimulated with hydrostatic pressure (Fig. 4d). Consistently, we also observed a reorganization of the actin cytoskeleton near the cell–cell junctions (Fig. 4e−g). Actin fibres, which were mainly parallel to the junctions in control conditions, were oriented perpendicularly to the cell–cell junctions under pressure stimulation. This latter feature is also a characteristic of remodeling adhesions that confirmed a reduction of junctional stability as previously reported[51,52]. Endothelial cells express two members of the classical cadherin family, VE-Cadherin and N-Cadherin[53], and their respective localization at adherens junctions changes as the monolayer matures[53–55]. Despite their high sequence homology, VE-Cadherin and N-Cadherin display different downstream signaling responses to stimuli in endothelial cells[53]. Moreover, endothelial cells shift to express different cadherin isoforms through a process that is called cadherin switching[56], which affects the phenotype and behavior of endothelial cells. Cadherin switching is one of the hallmarks of EndMT, a complex biological process in which endothelial cells acquire a mesenchymal phenotype characterized by increased proliferation, motility, and reduced barrier function[57].

Since partial EndMT is a potent inducer of angiogenesis[58,59], we investigated whether, besides the structural changes in the junctional morphology (Fig. 4c−g), hydrostatic pressure stimuli also affected cadherin levels. In control monolayers, VE-Cadherin was localized at adherens junctions, and only low amounts of N-Cadherin were detected, which mainly localized at the cell membrane. 24 h pressure stimulation resulted in a more junctional localization of N-Cadherin cadherin (Fig. 5a). A time course analysis of cadherin expression revealed that N-Cadherin was significantly increased after 24 h stimulation with 55 mmHg hydrostatic pressure and returned to basal level after 48 h (Fig. 5c, d). Samples under 20 mmHg had slightly increased VE-Cadherin levels after 2 h and 24 h, which returned to basal levels after 48 h. Conversely, under 55 mmHg pressure stimulation, a modest reduction of VE-Cadherin expression was observed at 2 h. At the mRNA level, an increase in both *CDH2* (encoding N-Cadherin) and *CDH5* (encoding VE-Cadherin) gene expression was detected at 2 h of 55 mmHg hydrostatic pressure, but there was no more difference at the later time points (Fig. 5e). Although we did not observe a complete cadherin-switch as suggested by the very mild down-regulation VE-Cadherin, the transient differential cadherin expression that we detected was pressure-magnitude dependent. We found a more pronounced effect of the arteriolar 55 mmHg stimulation compared to the capillary 20 mmHg (Figs. 3–4). Although not all the differences in cadherin expression were statistically significant, these variations, combined with the differences detected in the structural organization of the junctions, correlated with an overall weakening of the cell-cell junctions that was confirmed by the analysis of the monolayer barrier properties (Fig. 4). All these data suggest that the dynamic structural alterations of the adherens junctions are involved in hydrostatic pressure-mediated mechanotransduction. The pressure-induced differential junctional plasticity could play a role in a partial EndMT program, contributing to the increased angiogenesis that we observed under capillary pressure (Fig. 2).

## Hydrostatic pressure stimulation triggers YAP expression, nuclear localization, and signaling

Endothelial physiology is tightly regulated via cell-cell junctional complexes, which serve as a signaling hub that triggers intracellular responses[40]. Specifically, the dynamic remodeling of the cadherin complex can contribute to modulate expression, intracellular localization, and activity of several transcription factors[40,53,60–63]. Among them, the transcriptional co-factors β-Catenin and YAP, a known mediator of mechanotransduction[64], are known promoters of EndMT and angiogenesis[65–68].

We thus evaluated if β-Catenin and YAP nuclear localization and activity are differentially regulated under pressure. At 24 h of pressure stimulation, β-Catenin immunostaining revealed a fuzzy and more dispersed

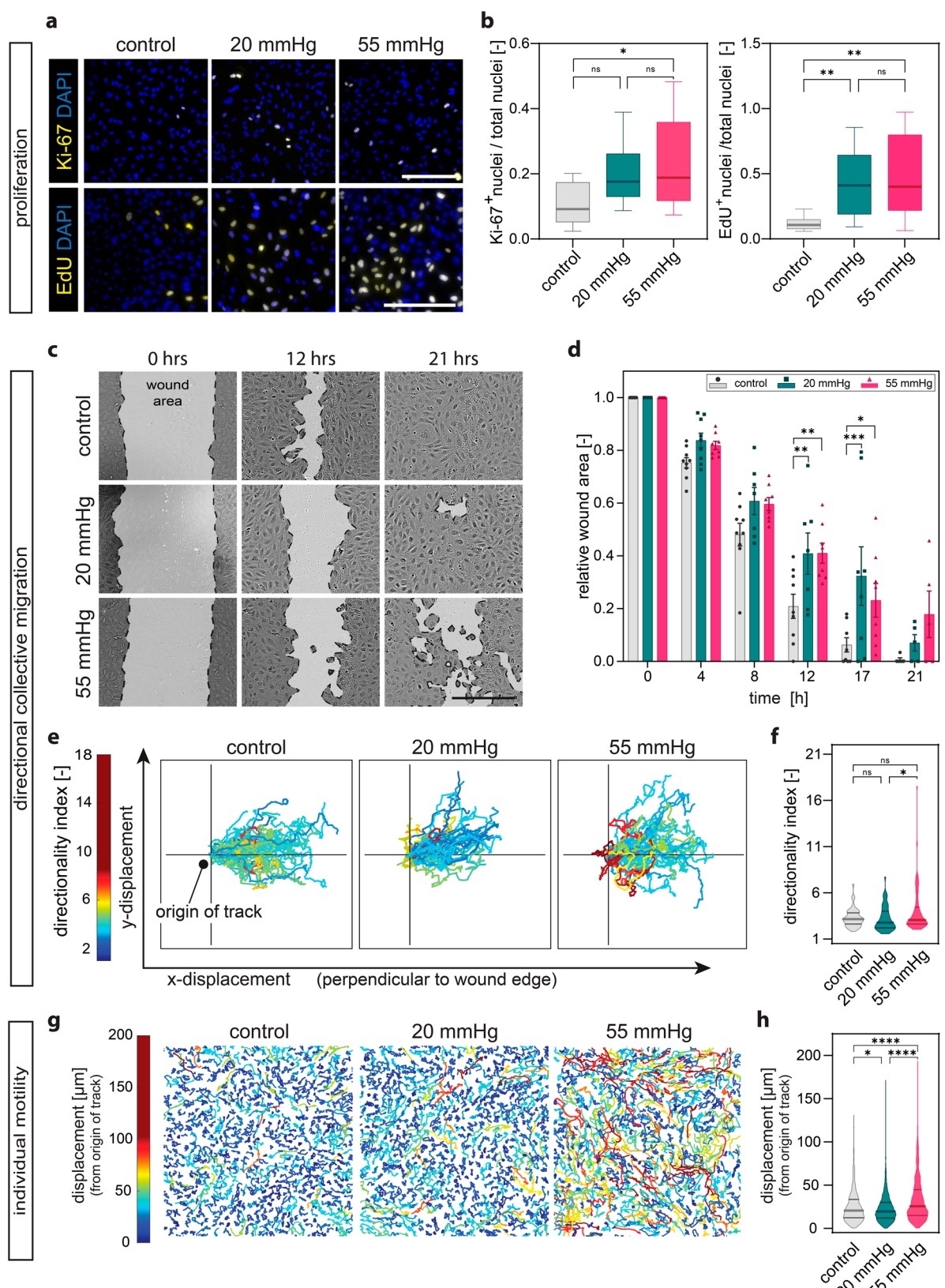

cytoplasmic localization (especially in the 55 mmHg condition) (Fig. S1a). However, Western blot analysis revealed comparable levels of phosphorylated β-Catenin (p-β-Catenin) (Fig. S1b, c), also known as active β-Catenin. The phosphorylation at serine 552 is a crucial post-translational modification that affects the stability and activity of β-Catenin[69,70]. Phosphorylation

of glycogen synthase kinase-3β (p-GSK3β), which regulates β-Catenin signaling activity[71], was also comparable between the samples (Fig. S1d, e). Furthermore, expression of the gene encoding axis inhibition protein 2 (*AXIN2*) (Fig. S1f), a β-Catenin target gene, was not significantly affected. AXIN2 acts in a negative feedback loop to limit and fine-tune β-Catenin

**Fig. 3 | Proliferation, directional collective migration, and individual motility are modified under microvascular hydrostatic pressure. a** Representative confocal z-projection images of HUVEC monolayers after 24 h under pressure. Ki-67- (upper panels) and EdU-positive-nuclei (lower panel) stain proliferating cells. DAPI counterstains the total number of nuclei. **b** Quantification of Ki-67- (left panel) and EdU-positive nuclei (right panel) in HUVEC monolayers after 24 h under pressure, $N = 3$ independent experiments, $n \geq 9$ analysed fields of view per group. **c** Representative light microscopy images of wound healing assay over time after pressure application. **d** Quantification of wounded area over 21 h of assay. $N = 3$ independent experiments, $n = 2-3$ analysed fields of view per group. **e** Tracks of the trajectories of individual cells at the wound leading edge over 21 h. **f** Quantification

of directionality index. $N = 3$ independent experiments, $n \geq 43$ analysed tracks per condition pooled from 3 wounds per group. **g** Tracks of individual cell trajectories within mature monolayers over 22 h. **h** Quantification of cell displacement of individual tracks. $N = 3$ independent experiments, $n \geq 1230$ analysed tracks per condition. Data information: (**b**) Graphs show median and 25th to 75th percentile, whiskers indicate min and max values. ns = non-significant; *$p > 0.05$, **$p < 0.01$. Kruskal-Wallis test; (**d**) Graphs show mean ± s.e.m. *$p < 0.05$, **$p < 0.01$, ***$p < 0.001$; for each timepoint two-way ANOVA with Tukey's post-hoc test. **f, h** Lines in violin plots show median and quartiles. *$p < 0.05$, ****$p < 0.0001$; Kruskal-Wallis test. Scale bar (**a**): 300 μm, (**c**): 400 μm.

signaling. These results argue against a difference in β-Catenin signaling between pressure-stimulated samples and control. Conversely, after 24 h of stimulation, we observed a pressure magnitude-dependent strong YAP nuclear shuttling, as shown by immunostaining (Fig. 6a–c) and nuclear/cytoplasmic fractionation assay followed by Western blot (Fig. 6d, e). Moreover, a slight reduction in YAP junctional localization was detected (Fig. 6b)[62,72].

YAP-modulated endothelial cell functions are regulated by various mechanisms that modulate signaling pathways such as the Hippo and the vascular endothelial growth factor (VEGF) pathways, as well as actin cytoskeleton dynamics. These include cell-cell contact inhibition[62,63,72], mechanical forces such as shear stress[73], and stiffness of the microenvironment[74], which in vivo in physiological conditions is more compliant and softer[75]. We thus evaluated if pressurization of cells cultured on soft collagen hydrogels (~3−4 kPa stiffness[76]) also triggers YAP nuclear relocalization. Also, in endothelial cells seeded on softer substrates, which acquired an elongated shape as previously reported[74], YAP shuttled into the nucleus upon pressure stimulation (Fig. S2a, b), although the phenotype appeared less severe.

It has previously been reported that hydrostatic pressure increases *YAP1* gene expression in a mouse model of unilateral pneumonectomy[29]. A time course analysis of *YAP1* expression revealed a mild increase already after 2 h of 55 mmHg pressure stimulation, while a strong and significant increase in *YAP1* gene expression was detected after 24 h of both 20 and 55 mmHg stimulation compared to the control (Fig. 6f−h). Expression of the YAP target gene ankyrin repeat domain 1 (*ANKRD1*)[65] was also increased in pressurized samples compared to control (Fig. 6i).

To exclude that the use of the custom pressure bioreactor deprived the cell cultures of adequate oxygen supply and eventually altered the cell behavior, we compared the gene expression of 2D classic cell culture (petri dish) and 2D 0 mmHg bioreactor cell culture (referred to as "control" in this study). As shown in Fig. S3a, the bioreactor 2D culture had no effect on the classical hypoxia-regulated genes carbonic anhydrase 9 (*CA9*) and prolyl-4-hydroxylase domain 3 (*PHD3*)[77], but increased the expression of glucose transporter 1 (*GLUT1*). GLUT1, besides being a hypoxia target, plays important roles in cells forming blood-tissue barriers such as endothelial cells and astrocytes[78]. Moreover, there were no differences in the expression of YAP (mRNA and protein level) and its target genes (mRNA level) (Fig. S3b, c) as well as in YAP nuclear relocalization (Fig. S3d).

Altogether, these data suggest that hydrostatic pressure stimulation triggers YAP activation and signaling through its translocation into the nucleus, or by inducing its transcription or both. The transcriptional regulation of *YAP1* by hydrostatic pressure is consistent with previously reported data[29].

## Hydrostatic pressure stimulation contributes to partial EndMT and angiogenesis

Partial EndMT has been described as the differentiation into cells with intermediate endothelial and mesenchymal features that are beneficial for sprouting angiogenesis[59]. It was previously reported by our group and others that YAP positively regulates EndMT and sprouting angiogenesis in multiple ways[68,79], including the expression of key EndMT markers[80,81]. We thus evaluated the expression of EndMT-driving transcription factors and

mesenchymal markers. Besides the transiently increased expression of *CDH2* (Fig.5e) and of the genes encoding the angiogenic inducers ANKRD1[65,82] (Fig. 6i), fibroblast growth factor 2 (FGF2)[83], and VEGFA and its receptor VEGF receptor 2 (VEGFR2)[84] (Fig. 7a), we also detected a transiently increased expression of α-smooth muscle actin (*ACTA2*) at 55 mmHg[68]. Moreover, we found increased expression of zinc finger E-box binding homeobox 2 (*ZEB2*) and receptor-regulated SMAD family member 2 (*SMAD2*)[85], two of the early transcription factors involved in EndMT, under these conditions (Fig. 7b). Of note, only samples under capillary 20 mmHg, but not 55 mmHg, significantly upregulated the pro-angiogenic genes *VEGFA* and *VEGFR2* (Fig. 7a)[84]. Simultaneously, we did not detect increased expression of genes encoding EndMT early transcription factors under capillary pressure (Fig. 7b), in agreement with the hypothesis that 20 and 55 mmHg do not activate the same intracellular signaling. Consistent with our previous results, these increases were also detected at the early time point (2 h), indicative of a dynamic adaptive response of endothelial cells to pressure. Together, our data suggest that YAP may act as an upstream regulator of the partial EndMT and angiogenesis programs induced by static monolayer pressurization.

## Inhibition of YAP transcriptional activity impairs pressure-triggered sprouting angiogenesis

To verify this hypothesis, we treated pressurized endothelial monolayers with verteporfin (VP), a pharmacological inhibitor of YAP transcriptional activity[86,87]. Figure 8a shows that VP led to reduced expression of the classical YAP target genes *CDH2, CYR61* and *ANKRD1* in both pressurized and control samples, compared to their vehicle (DMSO)-treated counterparts. To evaluate the effect of VP on hydrostatic pressure-dependent angiogenesis, we treated a 3D in vitro organotypic vessel model of HUVECs, as we used this cell type to identify the role of YAP in mediating hydrostatic pressure mechanoresponses in vitro. As shown in Fig. 8b, c and in line with what was previously observed in the aortic ring assay (Fig. 2), 20 mmHg simulation increased angiogenic sprouting compared to the control condition of no pressure. Of note, also in this in vitro model, 20 mmHg stimulation was more efficient in inducing sprouting angiogenesis than 55 mmHg. For this reason, we coupled VP treatment with 20 mmHg hydrostatic pressure stimulation for 48 h and analysed the sprouting angiogenesis. Inhibition of YAP transcriptional activity indeed significantly impaired sprouting angiogenesis triggered by cell pressurization (Fig. 8b, c). Furthermore, VP treatment also reduced sprouting at 20 mmHg in the ex vivo aortic ring assay (Fig. 8d, e).

Overall, our results strongly suggest that hydrostatic pressure promotes angiogenesis through YAP expression, intracellular localization, and signaling.

## Hydrostatic pressure stimulation activates YAP signaling in different types of endothelial cells

Mechanical signals (e.g., wall shear stress) contribute to sustain endothelial phenotype heterogeneity[88], but they are also the major modulators of endothelial plasticity, being able to modify and even switch phenotypes and functions of primary endothelial cells from different vascular beds in vivo and in vitro[70,89–93]. We therefore investigated if the pressure-induced YAP activation and signaling observed in HUVECs also occurs in other types of

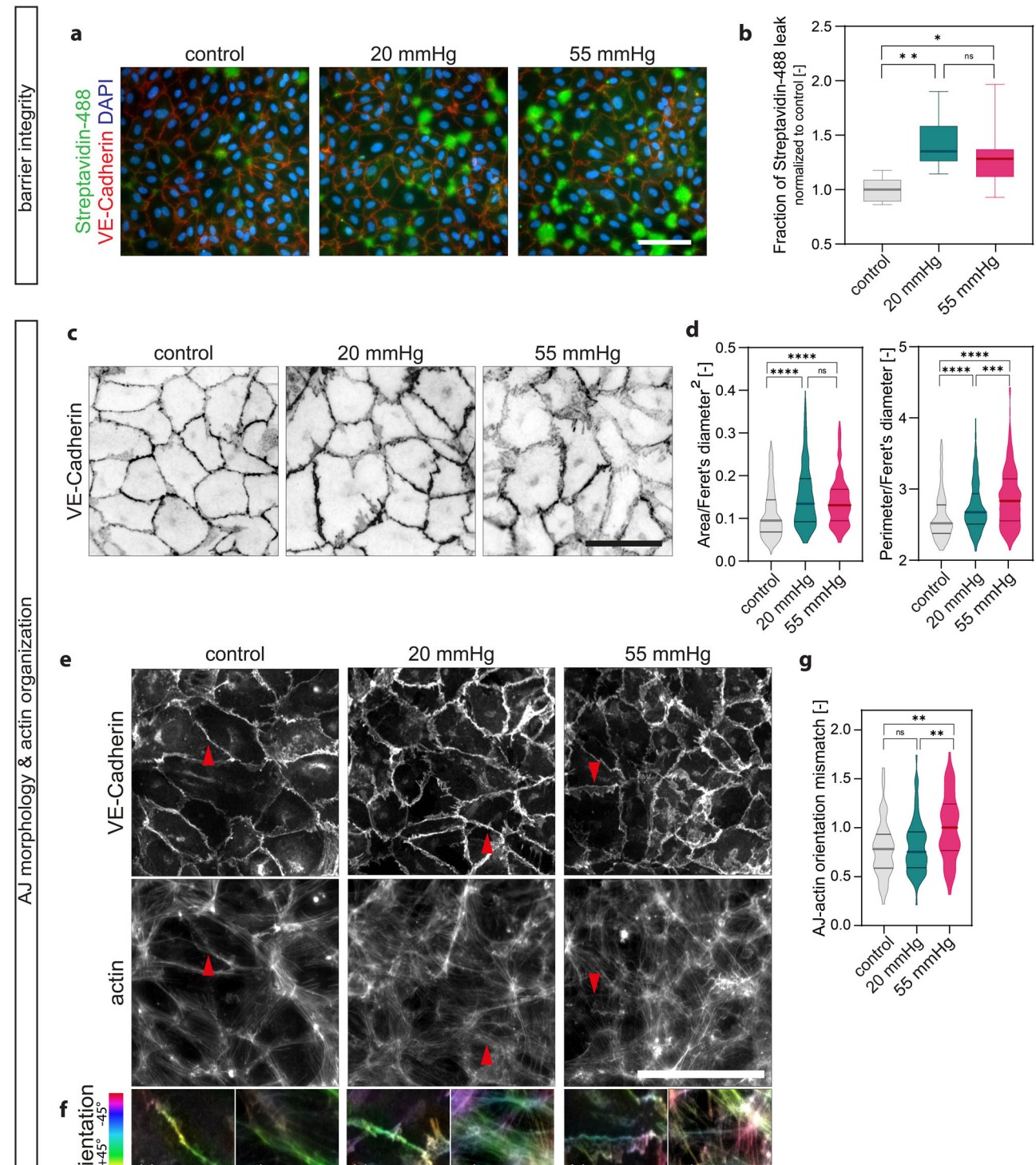

**Fig. 4 | Monolayer integrity and junctional organization under hydrostatic pressure stimulation. a** Representative immunofluorescence z-projection images of HUVEC monolayers after 24 h 54 showing barrier properties. Green area is the biotin-streptavidin leak, VE-Cadherin (red), Streptavidin-488 (green), DAPI (blue). **b** Quantification of Biotin-Streptavidin assay, *N* = 4 independent experiments per group, *n* = 3 analysed fields of view per condition and experiment. **c** Inverted representative confocal z-projection images of HUVEC monolayers showing adherens junction morphology (VE-Cadherin). **d** Shape descriptors of adherens junctions (measured from vertex to vertex normalized by longest distance in the shape (Feret's diameter)). *N* = 3 independent experiments per group, *n* ≥ 248 analysed adherens junctions per condition pooled from 3 fields of view per condition and experiment. **e** Immunofluorescence z-projection images of HUVEC monolayers

after 24 h showing adherens junction (VE-Cadherin) and actin cytoskeleton organization. Red arrows indicate adherens junctions represented in (**f**). **f** Representative z-projection images of adherens junction (left) and actin (right) colored according to orientation (see Methods). **g** Quantification of orientation mismatch of adherens junctions compared to the actin cytoskeleton. *N* = 3 independent experiments per group, *n* = 56−76 analysed adherens junctions per condition from 3 fields of view per condition and experiment. Data information: **b** Graphs show median and 25th to 75th percentile, whiskers indicate min and max values, ns = non-significant, **p* < 0.05, ***p* < 0.01; Kruskal-Wallis test, Dunn's post-hoc. **d, g** Lines in violin plots show median and quartiles. ns = non-significant; ***p* < 0.01, ****p* < 0.001, *****p* < 0.0001; Kruskal-Wallis test, Dunn's post-hoc. Scale bars: 100 μm. AJ: adherens junction.

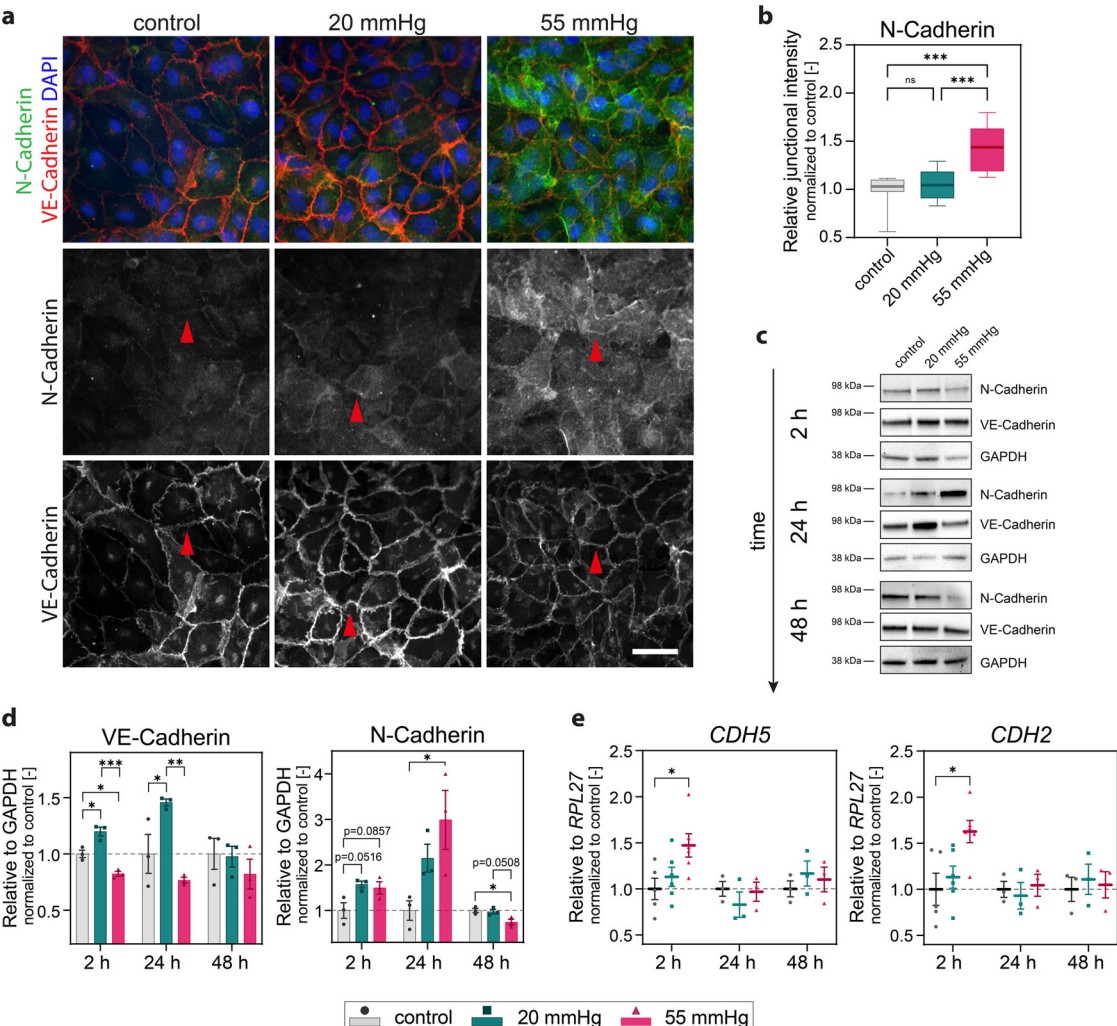

**Fig. 5 | Hydrostatic pressure stimulation triggers differential cadherin expression. a** Representative widefield z-projection images of monolayers showing cadherin localization. VE-Cadherin (red), N-Cadherin (green), DAPI (blue). Red arrowheads highlight junctions. **b** Quantification of N-Cadherin fluorescence intensity localized at adherens junctions relative to background after 24 h under pressure. $N = 3$ independent experiments per group, $n = 3–4$ analysed fields of view per condition and experiment. **c** Western blot analysis of N-Cadherin and VE-Cadherin after exposure to pressure for 2 h, 24 h, and 48 h, glyceraldehyde-3-phosphate dehydrogenase (GAPDH) is the loading control. **d** Densitometry of N-Cadherin and VE-Cadherin protein abundance relative to GAPDH. $N = 3$ independent experiments per group. **e** RT-qPCR gene expression analysis of CDH2 (encoding N-Cadherin) and CDH5 (encoding VE-Cadherin) relative to RPL27. $N \geq 3$ independent experiments per group. Data information: (**b**) Graphs show median and 25th to 75th percentile, whiskers indicate min and max values. ns = not significant; ***$p < 0.001$. Kruskal-Wallis test; (**d**) Graphs show mean ± s.e.m. Reported are multiplicity adjusted $p < 0.1$, *$p < 0.05$, **$p < 0.01$, ***$p < 0.001$; for each timepoint two-way ANOVA with Tukey's post-hoc test.; (**e**) Graphs show mean ± s.e.m. *$p < 0.05$; for each timepoint one-way ANOVA with Dunnett's post-hoc test. Scale bar: 100 μm.

endothelial cells. To this aim, we pressurized human aortic endothelial cells (HAoECs), and we observed a similar response. In line with what was observed in HUVECs, we found a mild, but non-significant increase in the protein levels of N-Cadherin (Fig. S4a, b) and YAP (Fig. S4c, d) in HAoECs under pressure compared to control. In addition, the nuclear localization of YAP was promoted by pressure stimulation (Fig. S4e, f). 20 mmHg pressure stimulation also increased the expression of connective tissue growth factor (*CTGF*) and inhibin subunit beta A (*INHBA*) (Fig. S4g), which are known YAP target genes[62,68]. Last, the 3D in vitro organotypic vessel model of HAoECs recapitulated the sprouting angiogenesis observed for HUVECs. As shown in Fig. S4h, i, 20 mmHg pressure simulation strongly promoted angiogenic sprouting, while VP treatment counteracted the effect of hydrostatic pressure.

Taken together, these results suggest that the mechanotransduction responses induced by hydrostatic pressure stimulation trigger similar effects in different types of vascular endothelial cells.

## Discussion

This work demonstrates that static stimulation of endothelial cells from different vascular beds with a hydrostatic pressure value characteristic of capillaries, induces partial EndMT and sprouting angiogenesis and that these cellular responses are dependent on YAP signaling. So far, there was limited evidence for a link between hydrostatic pressure and angiogenesis in vivo or in vitro, with few reports suggesting that it plays a secondary role compared to other mechanical stimuli such as shear stress and deformation[1]. Our work provides further insights into the mechanical regulation of angiogenesis, an important endothelial process with a key role in physiology and pathology. We show the effects of pressure at different levels - from molecular, cellular to behavioral processes - proposing hydrostatic pressure as a contributing factor in the control of angiogenesis.

Consistent with the current knowledge, we found that hydrostatic pressure induced proliferation of endothelial cells[22,25,94] and impaired the proper organization of adherens junctions[2,27]. We also showed that

**Fig. 6 | Hydrostatic pressure stimulation induces YAP expression, nuclear localization, and signaling. a** Representative widefield z-projection images of monolayers after 24 h under pressure showing YAP intracellular localization. VE-Cadherin (red), YAP (green), DAPI (blue). **b** Quantification of YAP localization at adherens junctions after 24 h of stimulation. *N* = 3 independent experiments per group, *n* ≥ 2 analysed fields of view per condition and experiment. **c** Quantification of YAP nuclear localization after 24 h of stimulation. *N* = 3 independent experiments per group, *n* ≥ 407 analysed nuclei pooled from ≥3 fields of view per condition and experiment. **d** Western blot for YAP nuclear-cytoplasmic fractionation in endothelial monolayers after exposure to pressure for 24 h. Lamin B1 and GAPDH are nuclear and cytoplasmic loading controls, respectively. **e** Densitometry of YAP protein abundance relative to Lamin B1 and GAPDH. *N* = 6 independent experiments per group. **f** Western blot for YAP expression in endothelial monolayers after exposure to pressure for 2 h, 24 h, and 48 h, Vinculin is the loading control. **g** Densitometry of YAP protein abundance relative to Vinculin. *N* ≥ 3 independent experiments per group. **h** RT-qPCR gene expression analysis of *YAP* relative to ribosomal protein L27 (*RPL27*). *N* = 3 independent experiments per group (except for 2 h 20 mmHg group). **i** RT-qPCR gene expression analysis of the YAP target genes connective tissue growth factor (*CTGF*), cysteine-rich angiogenic inducer 61 (*CYR61*) and ankyrin repeat domain containing 1 (*ANKRD1*) relative to *RPL27*. *N* ≥ 3 independent experiments per group. Data information: (**b**, **c**) Graphs show median and 25th to 75th percentile, whiskers indicate min and max values. ns = non-significant, *p < 0.05, ****p < 0.0001; Kruskal–Wallis test; (**e**) Graphs show mean ± s.e.m., ***p < 0.001, ****p < 0.0001; two-way ANOVA with Tukey's post-hoc test. Data points are measurements obtained from multiple not saturated exposure times of the membrane. **g** Graphs show mean ± s.e.m., *p < 0.05; for each timepoint Friedman test. **h, i** Graphs show mean ± s.e.m. *p < 0.05; for each timepoint one-way ANOVA with Dunnett's post-hoc test. Scale bars: 100 μm.

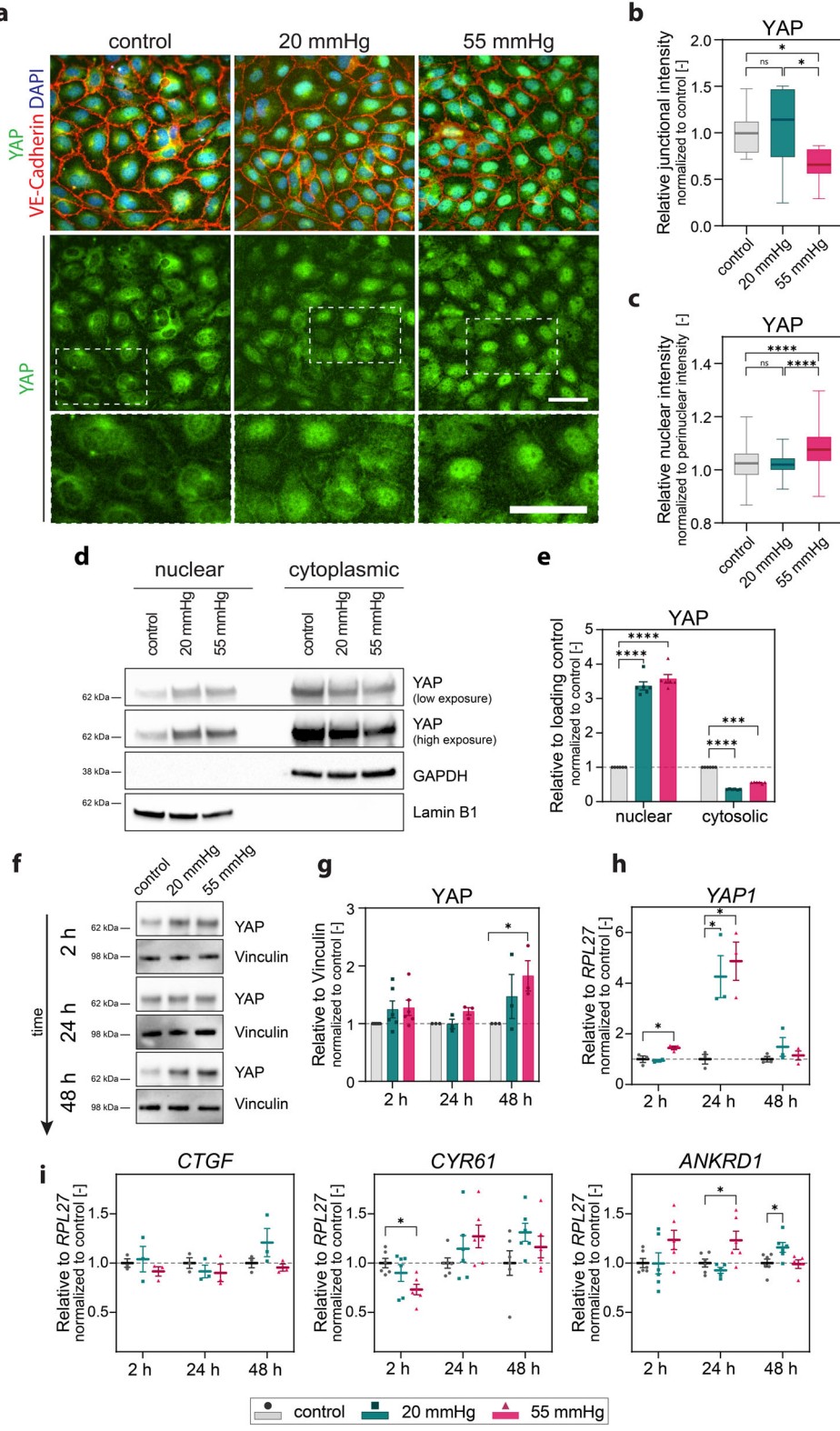

structural weakening of the adherens junctions[95] resulted in an increase in monolayer permeability as previously reported[26].

In addition, our findings highlight a hydrostatic pressure-driven contribution to endothelial cell migration. We show that endothelial cells are mechanosensitive to continuous capillary levels of hydrostatic pressure (20 mmHg)[15], which promoted both individual cell motility and directional collective migration. This, in combination with the increased proliferation

and monolayer structural weakening, positively contributed to sprouting angiogenesis. Of interest, our work also highlights how endothelial monolayers react to hydrostatic pressure stimuli to guarantee specific magnitude and context-dependent responses. Contrary to what was observed at 20 mmHg, the stimulation with static pressure of 55 mmHg resulted in defective sprouting. Interestingly, 55 mmHg is characteristic of terminal arteriole pressure where sprouting angiogenesis does not typically occur

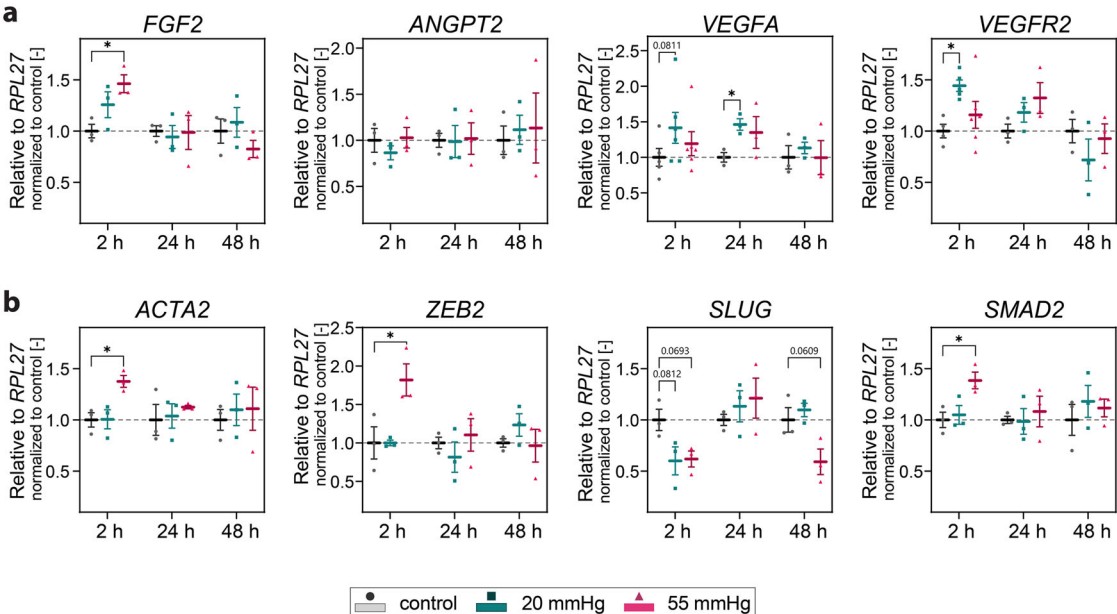

**Fig. 7 | Hydrostatic pressure triggers expression dynamics of angiogenic and EndMT markers. a** RT-qPCR gene expression analysis of pro-angiogenic markers relative to *RPL27*. $N \geq 3$ independent experiments per group. **b** RT-qPCR gene expression analysis of EndMT transcriptional signature relative to *RPL27*. $N = 3$ independent experiments per group. Data information: (**a**, **b**) Graphs show mean ± s.e.m. Reported are multiplicity adjusted $p < 0.1$, *$p < 0.05$; for each time-point one-way ANOVA with Dunnett's post-hoc test; fibroblast growth factor 2 (*FGF2*), vascular endothelial growth factor A (*VEGFA)*: for each timepoint Kruskal-Wallis test.

in vivo[13]. We detected that 55 mmHg stimulation promoted cell proliferation, individual cell motility within the monolayer, junctional dismantling, and impaired barrier function. However, it also strongly impeded collective directional migration, which in turn resulted in an overall reduced sprouting angiogenesis both in vitro and ex vivo. The latter could be the result of an incorrect balance of proliferation and migration during sprout formation[96].

We also found that molecular changes occurring in response to hydrostatic pressure were transient. Upon pressurization, the structural organization of the adherens junctions, which act as mechanosensors and mechanotransducers of pressure stimulation, was altered[46]. N-Cadherin, which in endothelial monolayers is typically localized at the cell membrane, was upregulated, and localized to cell-cell contacts. It was previously reported that N-Cadherin junctional localization in endothelial monolayers induces cell features characteristic of sprouting angiogenesis, such as increased proliferation, migration[53], and permeability in vitro[97] and in vivo[98]. We also showed that hydrostatic pressure reorganized adherens junctions into remodeling adhesions[40] with perpendicular actin fibres[51,52,99], indicating reduced junctional stabilization[47,100]. Cadherin switching is one of the hallmarks of complete EndMT[57], a process that generates endothelial-derived mesenchymal cells. The maintenance of junctional contacts and the concomitant acquisition of pro-proliferative and pro-migratory phenotypes here detected are instead the characteristics of a partial EndMT event. Partial EndMT, which generates cells with intermediate endothelial and mesenchymal features, is a process that activates sprouting angiogenesis in a transient and reversible way[59,101]. In agreement, in our context we found a transient activation of YAP, a key player in driving these angiogenic cellular responses. A recent paper reported that hydrostatic pressure induced YAP expression and activity in a model of mouse lung regeneration[29]. In our work we showed that monolayer pressurization with a capillary value of hydrostatic pressure activated YAP expression, nuclear shuttling, and transcriptional activity. Of interest and in line with previous literature[64], this phenotype was tuned by the stiffness of the substrate.

The reciprocal relationship between adherens junction remodeling and YAP activity has already been described in endothelial cells. YAP increases the turnover of VE-Cadherin at junctions, promoting endothelial cell migration[72], while YAP localization and activity are regulated by the

dynamic of VE-Cadherin-mediated adhesions[62,63]. In cancer cells, epithelial-to-mesenchymal transition is regulated by YAP itself, affecting the expression level of E- and N-cadherin[102]. A similar positive feedback loop could be present in our system, where YAP expression and signaling can contribute to the changes of junctional dynamics, thereby amplifying YAP-induced signals triggering partial EndMT and angiogenesis. Our results showed that the activation of this pathway is time-dependent, with transient YAP signaling occurring in the early period after static cell pressurization. Lastly, we showed that pharmacological inhibition of YAP activity was able to prevent hydrostatic pressure-induced sprouting angiogenesis. Endothelial cells in vivo possess remarkable phenotypic variability depending on the vascular bed and organs where they are located, and mechanical signals contribute to the heterogeneity of endothelial phenotypes[88]. Literature also reports about the role of mechanical forces as modulators of the endothelial plasticity, defined as the remarkable ability to change phenotypic characteristics and functions to adapt to various stimuli, at molecular, cellular and tissue levels[89]. We were interested in investigating the role of hydrostatic pressure in this context. We observed that hydrostatic pressure stimulation similarly modulated the mechanoresponses of endothelial cells of venous and arterial origin, activating YAP and inducing sprouting angiogenesis. Altogether, these results are in line with in vivo evidence supporting the concept that endothelial functions are mostly determined by the biomechanical properties of the environment[89–92]. In vivo, pathophysiological angiogenesis is triggered and regulated by a complex interplay between pro-angiogenic and anti-angiogenic biochemical and biophysical cues[7]. With this work, we demonstrate that independent of other stimuli present during angiogenesis, hydrostatic pressure alone, applied without transendothelial pressure gradient, pressure-fluctuations, or osmotic counterpart, can trigger cellular responses that are integrated dynamically towards a new homeostatic cellular phenotype. The crosstalk between signaling induced by multiple mechanical stimuli and by bioactive molecules could have both synergistic or antagonistic effects in regulating the same cellular process. This could explain why, although with less efficacy than the 20 mmHg, also the 55 mmHg stimulation alone induced a certain degree of sprouting. Although here we report about a single mechanical stimulation with the limitations that it does not reflect the complex and variable physiological

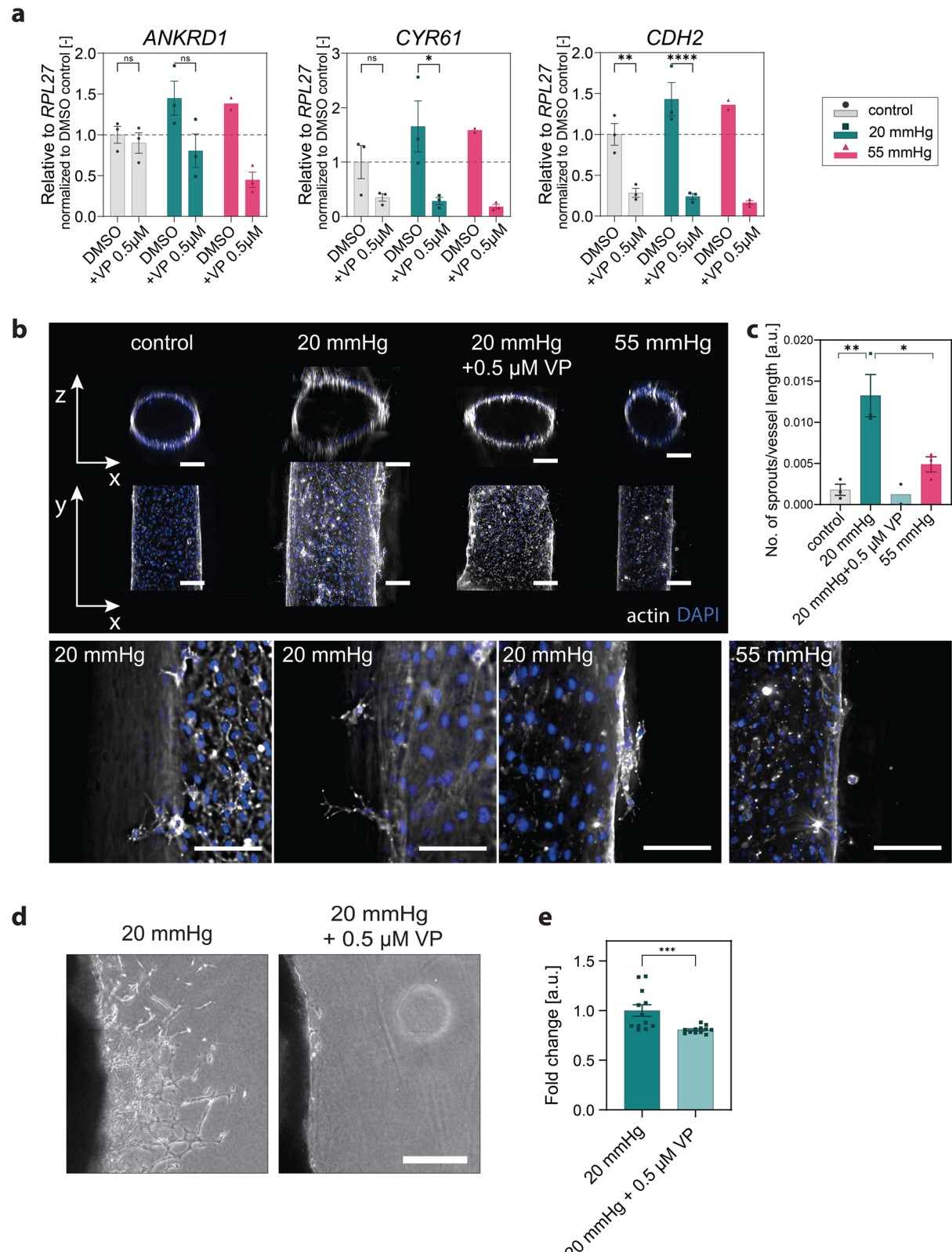

conditions in vivo, and we describe effects of pressure application on previously unpressurized (adapted to standard cell culture) cells in vitro, the employed pressure levels in this study are informed by realistic physiological blood pressure values.

Given the central role played by the vasculature in both physiological and pathological conditions, it is key to deeply understand and control the effect of blood pressure on endothelial homeostasis, junctional stability, and barrier function. Hydrostatic pressure may be relevant to maintain the physiological phenotype of endothelial cells in vitro and it could be a parameter integrated in cell culturing protocols. In the future, pressure stimulation could offer a mechanical alternative to pharmacological manipulation in biomedical applications. Moreover, our findings open new prospects in tissue-engineering settings as hydrostatic pressure can control endothelial physiology and ultimately angiogenesis.

**Fig. 8 | Hydrostatic pressure stimulation promotes sprouting angiogenesis through YAP activity. a** RT-qPCR gene expression analysis of the YAP target genes *ANKRD1*, *CYR61* and *CDH2* in endothelial monolayers relative to *RPL27* after 24 h under pressure stimulation with and without 0.5 μM Verteporfin (VP). $N = 3$ independent experiments for all groups (except $N = 2$ for DMSO 55 mmHg). **b** Deconvoluted representative widefield projection images of 3D organotypic models after 48 h under pressure stimulation with and without 0.5 μM VP. Actin (gray), DAPI (blue). Representative magnifications show angiogenic sprouts visualized by altered brightness, contrast, and projected planes of 20 mmHg and 55 mmHg conditions. **c** Quantification of sprouts in 3D organotypic models after

48 h under pressure. $N = 3$ independent experiments per group (except $N = 2$ for VP group). **d** Light microscope images of mouse aortic rings stimulated for 5 days with hydrostatic pressure (20 mmHg) with and without 0.5 μM VP. **e** Quantification of sprouts of mouse aortic rings stimulated for 5 days with hydrostatic pressure (20 mmHg) with and without 0.5 μM VP. $N = 3$ independent experiments, $n = 4$ per group. Data information: (**a**) Graphs show mean ± s.e.m. ns = non- significant, $*p < 0.05$, $**p < 0.01$, $***p < 0.001$, $****p < 0.0001$; two-way ANOVA with Tukey's post-hoc test. **c** Graphs show mean ± s.e.m. $*p < 0.05$, $**p < 0.01$; one-way ANOVA with Tukey's post-hoc test. **e** Graphs show mean ± s.e.m. $***p < 0.001$; Mann-Whitney test. Scale bars: 200 μm.

## Experimental section/methods

### Cell culture
Primary Human Umbilical Vein Endothelial Cells (HUVECs, C-12203, Promocell) and Human Aortic Endothelial Cells (HAoECs, C-12271, Promocell) were cultured using Endothelial Cell Growth Medium (ECGM, C-22010, Promocell) supplemented with 20% fetal bovine serum (FBS, A3160802, Thermo Fisher Scientific) and maintained at 37 °C and 5% $CO_2$. Before each seeding, dishes and glass slides were coated with 1.5% weight/volume (w/v) gelatin (Millipore) in deionized water (autoclaved, pH-adjusted to 7.4 and sterile filtered; prepared weekly). The resulting gelatin solution was incubated on substrates at 37 °C for at least 1 h and aspirated shortly before cells were seeded. Cells were routinely detached using ice cold Accutase (Thermo Fisher scientific).

For experiments, cells were seeded on gelatin-coated dishes at $4 \times 10^4$ cells/cm² in ECGM + 20% FBS, left to attach for 24 h, then medium was exchanged for ECGM to mature the monolayers for another 48 h with HUVECs and another 72 hours for HAoECs. All experiments were run in confluent mature monolayer configuration. Cells were used for experiments at passage 3 to 5 after isolation.

### Pressure bioreactor
A custom set-up was built to apply hydrostatic pressure using a column of cell culture medium. Molds for o-rings were 3D-printed using polylactic acid filament (Prusa). Polydimethylsiloxane (PDMS, Sylgard 184) mixed at a 1:10 cross-linker to elastomer ratio was degassed and poured onto molds to be cured at 65 °C for 1.5 h. The resulting o-rings were punctured using 1-mm biopsy punches for access to pressure chamber. Polytetra-fluoroethylene (PTFE) tubing inserted though the punctures in the O-rings served as adaptor pieces for silicone tubing of arbitrary length. The o-rings were placed on the upper rim of commercially available cell culture dishes and covered with polymethylmethacrylate (PMMA) plates and held in place by stainless steel screws and nuts. The end of the silicon tubing was connected to a reservoir that was opened to atmospheric pressure via a PTFE filter membrane. The correct height of water level was adjusted manually via an in-house pressure sensor (validated via theoretical water column height). All temperature resistant components were autoclaved using a dry cycle held for 20 min at 120 °C. Temperature sensitive materials were placed in a 70% ethanol bath, followed by several washes in PBS. Before experiments, cell culture medium was assimilated to incubator conditions for at least 2 h in large cell culture flasks equipped with PTFE filters in bottle caps.

### Gene expression analysis
RNA was extracted using RNeasy Kit (74004, Qiagen) according to the manufacturer's protocol. RNA amount was quantified by Nanodrop, and equal amount of cDNA was synthesized with a High-Capacity cDNA Reverse Transcription Kit (4368814, Thermo Fisher Scientific) following the manufacturer's protocol. qRT-PCR was then performed using PowerTrack™ SYBR Green Master Mix (A46012, Thermo Fisher Scientific) in a QuantStudio 5 (A28135). Primers used to target transcripts of interest are listed in the supplementary information sheet (Table S1). Resulting Ct values were normalized to the Ct-value of the housekeeping gene *RPL27* and are expressed as $2^{-\Delta\Delta Ct}$ (normalized to the control sample).

### Nuclear-cytoplasmic fractionation
Confluent monolayers of endothelial cells were pressure-stimulated for 24 h and then nuclear and cytoplasmic protein extraction was performed with the NE-PER Nuclear and Cytoplasmic Extraction kit (Thermo Fisher Scientific) following the manufacturer's instructions. 6 biological independent samples/condition were pooled together to obtain sufficient nuclear protein extracts. After protein fractionation, samples were processed using western blotting.

### Western blotting
Immediately after removal of pressure, samples were washed once rapidly using Dulbecco's phosphate-buffered saline (with calcium, magnesium, DPBS, Thermo Fisher Scientific) before lysis. Whole cell protein lysates were collected using Sample Buffer (2% sodium dodecyl sulfate, 20% glycerol, and 125 mM Tris-HCl, pH 6.8) heated to 98 °C using vigorous scraping followed by several passes through a 26-gauge needle. After 10 min of boiling at 98 °C, samples were centrifuged at 13,300 *g* for 15 min. The upper phase was collected and stored at −20 °C until further use. For western blotting, equal amounts of total protein were loaded onto a Bolt™ 4 to 12%, Bis-Tris Gel (Thermo Fisher Scientific) and run at constant 150 Volt until sufficient separation was achieved. SeeBlue® Plus2 Pre-Stained Protein Standard (4−250 kDa range) (Thermo Fisher Scientific) was used as ladder. Proteins were transferred to iBlot™ 2 nitrocellulose membranes (Thermo Fisher Scientific) using the iBolt 2 system (Thermo Fisher Scientific). Unspecific binding sites on membranes were blocked using 10% bovine serum albumin (BSA; Sigma-Aldrich) in PBS for 1 h, before over-night incubation in primary antibody diluted in 5% BSA in PBS at 4 °C. Secondary antibodies were incubated at standard temperature for 1 h. Primary and secondary antibodies and employed dilutions are listed in the supplementary information sheet (Table S2). No less than three washing steps in between changing solutions were performed using washing buffer (0.002% volume/volume Tween20 in PBS). Bands were visualized using BIORAD ChemiDoc XRS⁺ machine and intensity was measured using ImageLab software.

### Verteporfin treatment
Verteporfin (5305, Tocris Bioscience) was dissolved in sterile DMSO and used at final concentration of 0.5 μM. Verteporfin and equivalent amounts of DMSO were supplemented to medium shortly before experiments were set up. Endothelial monolayers were treated with Verteporfin for 24 h, while 3D organotypic samples were treated with Verteporfin for 48 h. For the aortic ring assay with Verteporfin, medium was exchanged with newly prepared medium with or without Verteporfin every 48 h.

### 3D aortic ring assay
Post-mortem tissue of wild-type C57BL/6 mice (12 weeks post birth, female, $CO_2$ euthanised) were used. Mouse tissue could be obtained after the end of an independent mouse study, for which all relevant ethical regulations for animal use were followed. The primary use of the animals in experiment was ethically evaluated and authorized by the cantonal authority (Kantonales Veterinärmat Zürich) and licensed under number ZH 015/2022. The use of leftover tissue after ending a licensed animal study is in accordance with federal law and approved on the whole by the cantonal authorities – no separate authorization number is therefore annotated. We have complied with all relevant ethical regulations for animal use. Aorta was dissected from

the thoracic cavity from four animals, and immediately stored in ECGM supplemented with 20% FBS on ice. Aortas were placed on a shaker to gently rinse. 1.5 mm collagen type I (5 mg/mL, bovine Type 1 collagen "Acido soluble collagen solution", SYMATESE, ACI600) was deposited in petri dishes at a final concentration of 3.3 mg/mL to be incubated for 30 min at 37 °C. Thoracic aortas were cut into 1 mm long rings and left on the collagen substrate for 15 min at 37 °C before adding another layer of 3 mm collagen. Dishes were filled with 10 mL ECGM supplemented with 20% FBS and pressurized (in pressure bioreactor as described above) after curing the collagen for another 20 min at 37 °C. Pressure was applied immediately and maintained without media change for 4 days. Bioreactors were removed from incubators daily to image samples without interruption of pressure application. Obtained images were manually stitched and area measured using ImageJ.

### 3D organotypic vessel model

The developed fabrication method was obtained by combination and modifications of previous protocols[103,104]. Custom PDMS gaskets were manufactured using acetone bonded PMMA molds. PDMS gaskets were pierced using 1-mm biopsy punches to create in- and outlets and then plasma-bonded to microscope slides. Poly-L-lysine (0.1% w/v in water, Sigma-Aldrich) was injected into the gaskets and resulting models were autoclaved. Acupuncture needles (SEIRIN) were cleaned in de-ionized water using a sonicator, autoclaved and immersed in sterile filtered 5% weight/volume bovine serum albumin (BSA) in PBS. Collagen type I (5 mg/mL, bovine Type 1 collagen "Acid soluble collagen solution", SYMATESE, ACI600) was cast into PDMS gaskets on ice to a final concentration of 2.31 mg/mL, and the needle was immediately pushed through the inlet. For the first 5 min after casting, needles were gently rotated once every 30 s at room temperature, then models were incubated for another 30 min in a cell culture incubator. Tubing was filled with medium and immediately connected to inlet, and at least 500 μL of medium was carefully flowed through via pressure overhead. After overnight incubation, 20 μL of single cell suspension of HUVECs or HAoECs (2 million cells/mL) was injected into models, and tubing clamped immediately thereafter to avoid flow. Models were left upside-down (turned 180°) for 2 h after seeding to endothelialize the upper half of the channel. 24 h after seeding, samples were pressurized by releasing clamps on tubing and increasing the height of reservoirs to required pressure values.

### 2D cell culture on soft collagen gels

A 1-mm-thick layer of collagen type I (5 mg/mL, bovine Type 1 collagen "Acid soluble collagen solution", SYMATESE, ACI600), was cast onto petri dishes reaching a final concentration of 3.3 mg/mL, and then incubated at 37 °C for 30 min (~3−4 kPa stiffness[76]). HUVECs were seeded at the same densities and were cultured identical to the protocols described in "Cell Culture" above.

### EdU proliferation assay

HUVEC monolayers were matured as described above in "Cell Culture". Half of the medium was collected from dishes to dilute 5-ethynyl-2'-deoxyuridine (EdU, 10 mM in DMSO, base click) to a final concentration of 20 μM. The diluted EdU was incubated with cells at 37 °C for 45 min before fluid pressurization for 24 h. EdU was detected using ClickTech EdU Cell Proliferation Kit (BCK-EdU647IM100, baseclick) by following the manufacturer's instructions. Total number of nuclei were determined by counterstaining the samples with DAPI.

### Wound healing assay

Matured and confluent monolayers were scratched with a sterile 1000 μL pipette tip. Debris was washed off the wounded area with pre-warmed Dulbecco's phosphate-buffered saline (with calcium, magnesium, DPBS, Thermo Fisher Scientific), then samples were placed into a pressure bioreactor with fresh ECGM for 20−22 h, and live imaging analysis was

performed. Median intensity projection over time was subtracted from each acquired image to remove background.

### Paracellular barrier properties assay

Endothelial monolayers were cultured on biotin-conjugated 1.5% gelatin. After 24 h of pressure application as described above, the bioreactor was opened, and medium was exchanged for prewarmed ECGM with Oregon Green 488-conjugated avidin (Thermo Fisher Scientific) at a final concentration of 25 μg/mL for 1 min. Afterwards, the medium was removed, and the cells were fixed using ice-cold 100% methanol for 10 min. Subsequently, residual unbound avidin was eliminated by washing the samples with PBS after fixation[44].

### Live imaging and cell tracking

Cell migration was monitored using an inverted Nikon-Ti wide-field microscope (Nikon) within a controlled condition (temperature of 37 °C and $CO_2$ concentration of 5%) incubation chamber (Life Imaging Services) equipped with an Orca R-2 CCD camera (Hamamatsu Photonics). Images were collected with a 10X, 0.45 NA long-distance objective (Plan Fluor, Nikon). Time-lapse experiments were set to routinely collect images in different spatial positions of the sample, in the brightfield channel with a time resolution of 10 min.

For tracking of individual cell trajectories, TrackMate Plugin[105] in Fiji was used on timelapse-images obtained from NucBlue™ Live ReadyProbes™ Reagent (Hoechst 33342) stained nuclei. The dye was incubated with matured monolayers for 15 min according to the manufacturer's recommendations before being replaced by pre-warmed ECGM. Dishes were incorporated in a pressure bioreactor as described above, however, ECGM was supplemented with one drop of NucBlue™/10 mL medium. Tracking data imported from TrackMate were analysed and graphed using custom MATLAB (MathWorks) code. Single cells at the wound edge were manually tracked using the Fiji Manual Tracking plugin. The directionality index was computed for each cell by summing the distances migrated at every time step and dividing it by the distance between the initial and final positions.

### Immunofluorescence staining

Monolayers were fixed for 15 min with 4% formaldehyde (Thermo Fisher Scientific) in phosphate-buffered saline (PBS) at room temperature. Next, the samples were permeabilized with 0.5% Triton X-100 (Sigma-Aldrich) in PBS for 10 min. Afterwards, they were incubated in 5% w/v BSA (Sigma-Aldrich) in PBS for 45 min at room temperature. The samples were incubated with the respective primary antibodies overnight at 4 °C. Subsequently, they were rinsed three times for 5 min with PBS and then incubated with the corresponding secondary antibody for 1 h at room temperature. Finally, the samples were washed three times for 5 min with PBS. For staining of nuclei, DAPI and fluorescent phalloidin were added during a washing step.

Mouse aortic rings and 3D organotypic blood vessel models embedded in collagen were fixed for 30 min in 4% paraformaldehyde at room temperature, followed by a permeabilization step using 0.3% Triton X-100 in PBS for 30 min. Staining was performed as described above, however, incubation steps for 3D samples were increased to 24 h, and washing steps performed for at least 6 h while gently agitated on a shaker. Primary and secondary antibodies and employed dilutions are listed in the supplementary information sheet (Table S2).

### Image acquisition

Samples stained for immunofluorescence were imaged with an inverted Nikon-Ti spinning disk confocal microscope (Nikon) equipped with an Andor DU-888 camera (Oxford Instruments) and a pE-100 LED illumination system (CoolLED Ltd., Andover) using 60X (1.4 NA) oil immersion, 40X (0.75 NA) oil immersion or 20X (0.45 NA) extra long-distance) objectives (Plan Fluor, Nikon). To ensure comparability, images of the same antigen were acquired using constant acquisition settings.

## Image analysis

Custom-made codes were created in Fiji (National Institute of Health) for the image analysis of protein localization[106]. To identify adherens junctions in each field of view, a mask (adherens junctions mask) was generated using the VE-Cadherin signal: z-stacks of 50 μm height were projected using "Average Intensity" function, then "Subtract Background" function was employed (sliding rolling radius of 300). The resulting image was then converted to a binary image using the Fiji default algorithm. The N-Cadherin signal was processed as follows: z-stacks of 50 μm height were projected using "Average Intensity" function, then "Subtract Background" function was employed (slice rolling radius of 80). Adherens junction mask was multiplied with the processed N-Cadherin signal using "Image Calculator" function, and mean intensity in remaining pixels was measured. YAP junctional signal was processed as follows: z-stacks of 50 μm height were projected using "Maximum Intensity" function, then "Subtract Background" function was employed (slice rolling radius of 25). The resulting image was thresholded (set values for all pictures the same) and multiplied with the adherens junction mask using "Image Calculator" function to obtain a junctional YAP mask. For each field of view the area of the junctional YAP mask was divided by the area of the adherens junction mask (Fig. S5).

YAP nuclear to perinuclear (i.e., cytoplasmic) signal was determined using a MATLAB (MathWorks) code[106,107]. Briefly, nuclear intensity (obtained from DAPI signal) is divided by the signal intensity in the perinuclear region (obtained by expansion of the nuclear region).

Cell-to-cell junction morphology was assessed by manually selecting outlines of the VE-Cadherin signal and measuring perimeter, area and Feret's diameter (see supplementary information Fig. S6a).

Adherens junction to actin orientation mismatch was analysed using a custom code. In brief, single adherens junctions were manually selected using VE-Cadherin z-projected (sum of pixel intensities) images (as in Fig. S6a). Selection was enlarged by 3 μm and the resulting square region of interest for the actin channel was sharpened (ImageJ default "sharpen" algorithm) and the corresponding VE-Cadherin channel was blurred (Gaussian, sigma radius of 2 pixels). Resulting actin and VE-Cadherin images were analysed for distribution of orientation angles separately (local window tensor of 4 pixels, cubic spline gradient) using Orientation J Plugin in ImageJ[108]. Histograms were normalized to the total number of angles measured, and results combined into ten-degree bins (see supplementary information Fig. S6b, c). "AJ-actin orientation mismatch" is the sum of the absolute values of the difference for each bin between actin and VE-Cadherin orientation angles. As we performed immunostainings of actin and VE-Cadherin in all samples obtained from various types of experiments, for this analysis images from different experiments have been pooled together.

Z-stacks of 3D organotypic vessel models were deconvoluted using Huygens Remote Manager 3.9 (signal to noise ratio: 30,100; 70 iterations; quality change criterion 0.1; "cmle" deconvolution algorithm). Using IMARIS 9.6.0. software (Oxford Instruments), each model was 3D rotated and maximum intensity projected onto indicated planes.

## Statistics and reproducibility

Statistical analysis was performed using Graphpad Prism 9.4.1. software. Normality and heteroscedasticity for each group were assessed with Kolmogorov-Smirnov, and Brown-Forsythe and Bartlett's tests, respectively. ANOVA analysis was performed to compare groups that were normally distributed. No statistical methods were used to predetermine sample size. Unless stated elsewhere, all experiments were performed with at least three biological independent replicates in technical triplicates were performed. Statistical significance was set as follows: $*p < 0.05$, $**p < 0.01$, $***p < 0.001$ $****p < 0.0001$. Statistical tests and relative $p$ values are indicated in each figure legend.

## Data availability

The source data behind the graphs in the paper are provided in Supplementary Data 1 and 2. Uncropped western blot images are shown in Supplementary Fig. 7. All other data are available from the corresponding author [C.G.] on reasonable request.

## Code availability

The codes implemented for the above analyses are available open access in the ETH Zürich Research Collection with the following https://doi.org/10.3929/ethz-b-000682748.

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

## Acknowledgements

This work was supported by the Swiss National Science Foundation (grant SNFS; 205321_188828 to C.G.). The authors acknowledge Dr. Tobias

Schwarz of ScopeM (ETH Zürich) for his support & assistance in this work, and Dr. Céline Labouesse and Prof. Dr. Mark Tibbitt for the fruitful comments on the manuscript. E.M. and S.W. are members of the SKINTEGRITY.CH collaborative research program.

## Author contributions

Conceptualization: C.G., E.M.; Methodology: C.G., E.M.; Validation: D.A., D.R., A.A, P.H.; Formal analysis: D.A.; Investigation: D.A., A.A., D.R., P.H., D.Z.; Resources: C.G., S.W., N.C., V.F.; Data curation: D.A., C.G.; Writing - original draft: C.G., D.A.; Writing - review & editing: all authors; Supervision: C.G., E.M.; Project administration: C.G.; Funding acquisition: C.G.

## Competing interests

The authors declare no competing interests.
