## [Transparent Peer Review file · Communications Biology]

Hydrostatic pressure drives sprouting angiogenesis via adherens junction remodeling and YAP signaling

Corresponding Author: Dr Costanza Giampietro

Figures originally included in the author's rebuttal have been redacted from this file.

Version 0:

Reviewer comments:

Reviewer #1

(Remarks to the Author)

This manuscript demonstrates the mechanism by which hydrostatic pressure stimulates sprouting angiogenesis through EndMT characterized by endothelial cell proliferation and migration via YAP signaling. The topic is interesting and the manuscript uses various in vitro assays to delineate the mechanism, however the results look a little too preliminary and some figures and the interpretation are unclear. Detailed comments are below.

1. The authors use HUVECs in the study. Given the difference of pressure between arterial and vein, the significance of using HUVECs in this study is unclear.
2. Ki67 staining is used for evaluation of proliferation. Analysis of DNA synthesis using BrdU incorporation is helpful.
3. Cell-cell junction integrity image (Fig. 4a) and VE-cadherin image (Fig. 4c) are not consistent. Also the difference in junctional stability among pressure is not clear. Electron microscopy would show better junctional structure.
4. Changes in N-cadherin and VE-cadherin expression are unclear in Fig. 5, both cell staining and immunoblotting.
5. The authors demonstrated the YAP expression at the cell-cell junctional area. However, it is extremely faint in the staining in Fig.6 and cannot be compared between control and pressured cells.
6. In Fig.8, the authors used another assay to examine the effects of VP on endothelial sprouting. It is not clear why they switched the assay from the aortic ring assay.

Reviewer #2

(Remarks to the Author)

This study provides a comprehensive investigation into the influence of hydrostatic pressure on sprouting angiogenesis by examining various cellular functions using in vitro and ex vivo models. The tested cellular functions include proliferation, directional migration, motility, permeability, junctional stability, EndMT, and YAP expression. The findings demonstrate that hydrostatic pressure significantly affects sprouting angiogenesis through these cellular activities. This work sheds light on complex endothelial cell physiology under physiological-related hydrostatic pressure levels, which is significant to the angiogenesis field and beyond. It can be considered for publication after addressing the following comments.

1. As mentioned in this paper, other factors such as shear stress and growth factors play important roles in sprouting angiogenesis. The local mechanical environment could vary for angiogenesis such as in the wound healing process. To this end, have the co-effect of hydrostatic pressure and the other factors been considered, and would that potentially change conclusions presented in this work, for example, the conclusion that capillary hydrostatic pressure promotes sprouting while arteriole hydrostatic pressure inhibits it?
2. Presented in Figures 3 -5, hydrostatic pressure (statistical) significantly improved cell motility and the impact was pressure magnitude dependent. This is also demonstrated for cadherin switching. However, a higher (55 mmHg) hydrostatic pressure does not increase paracellular leakage, which does not match the observations mentioned above. This should be noted and discussed.
3. As discussed in the paragraph starting at line 213, no difference was identified with β -Catenin staining, but according to figure S1-a, the phospho- β -Catenin signal significantly increased under 55 mmHg after 24h. Please elaborate on the variation observed here.

4. In this study, an ex vivo 3D aortic ring assay, a 2D in vitro HUVECs model, and a 3D in vitro organotypic blood vessel model were used to test sprouting angiogenesis and different cellular activities. Have the variations between the models been assessed and shown to support a consistent line of research? For example, does hydrostatic pressure have the same impact on human v.s. mouse UVEC? Has any other assay been considered, for example, HDMEC sprouting, tube formation, or any in vivo assays?

Reviewer #3

(Remarks to the Author)

In this work Al-Nuaimi and co-authors study the impact of hydrostatic pressure on endothelial cells. They utilize mouse aortic endothelial cells, in 3D aortic ring assays, and human umbilical vein endothelial cells (HUVECs) to study angiogenic sprouting, proliferation, migration and junctional barrier integrity under hydrostatic pressure. The experiments are conducted under 0 mmHg (control), 20 mmHg (mean capillary pressure) and 55 mmHg (peripheral arteriole pressure). The authors further show that N-cadherin is transiently increased in HUVECs under induced hydrostatic pressure only after 24h and that YAP1 is localized to the nucleus. Accordingly, some YAP target genes are regulated under selected hydrostatic pressure conditions and Verteporfin treatment of HUVECs in a 3D organotypic model was able to restrict 20 mmHg-induced sprouting.

While the impact of hydrostatic pressure on endothelial cell functions, such as proliferation and junctional barrier maintenance has already been studied, the authors observe some additional new findings. The experiments are technically convincing and the manuscript is well written. However, this reviewer feels that the obtained data are insufficiently discussed and are to some extent overinterpreted. Selected additional experiments would strengthen the manuscript.

Major points:

1. While authors use arterial and venous endothelial cells, they do not analyze or discuss endothelial cell type differences throughout the manuscript. Aortic rings are dissected from mouse aortas, where the endothelial cells certainly experience higher hydrostatic pressure as the pressure values, they are subsequently subjected to in the aortic ring assay. Why do 55 mmHg show higher sprouting capacity than 0 mmHg controls? Should high pressures not prevent sprouting in aortic endothelial cells? In that context, how does venous capillary pressure massively increase sprouting in comparison to 0 mmHg controls?

The authors should perform key experiments, such as YAP nuclear localization, YAP target gene expression and 3D organotypic model experiments with human aortic endothelial cells to evaluate potential endothelial cell type differences.

2. Why aortic rings and 3D organotypic models do provide endothelial cells with a compliant/soft or more physiological stiffness environment, presumably standard endothelial cell culture dishes do not enable endothelial cells to accurately react to hydrostatic pressure. They authors should discuss this caveat as YAP/TAZ are regulated in endothelial cells via the stiffness microenvironment, which is obviously different between the here compared experimental approaches.

3. In line 82 the authors state that transient increased N-cadherin promote the weakening of adherens junctions. This statement is not confirmed in an experimental setup and simple is assumed based on correlation. Furthermore, "N-Cadherin, which in endothelial monolayers is typically localized at the cell membrane, was upregulated and strongly re-localized to cell-cell contacts." (Line 311-312). This statement cannot be seen from the images provided in Figure 5a. The Western blot in Figure 5c solely shows N-cadherin protein upregulation after 24h in the 55 mmHg-condition, which does not correlate with highest monolayer permeability under 20mmHg pressure in Figure 4a. Why does VE-cadherin not change on the protein level at all?

4. The transient nature of the in vitro phenotypes observed in this work should be addressed in more detail and related to potential in vivo situations. How long have the cells been cultured before they are subjected to pressure? What was the status of confluency?

Minor points:

1. In the supplemental data S2c the authors show that YAP is already increased after 48h in the bioreactor pressure control versus conventional cell culture pressure. Could the authors show what is the YAP expression after 24h in this experiment, as this is the usual time point studied in the main manuscript.

2. In Figure 6e ANKRD1 is not up-regulated after 24h with 20 mmHg but it is in Figure 8a. Can the authors explain the difference?

3. Figure 4a control and Figure 5a control show the same field of view. The authors should indicate that the information has been obtained from the same experiments.

4. Statements, such as "for the first time" (Line 35), should not be used.

Author Rebuttal letter:

COMMSBIO-23-4298: Rebuttal letter

A list of the references mentioned in our rebuttal letter is included at the end of this document.

Major modifications in the manuscript are highlighted in yellow.

Reviewers' comments:

Reviewer #1 (Remarks to the Author):

This manuscript demonstrates the mechanism by which hydrostatic pressure stimulates sprouting angiogenesis through EndMT characterized by endothelial cell proliferation and migration via YAP signaling. The topic is interesting, and the manuscript uses various in vitro assays to delineate the mechanism, however the results look a little too preliminary and some figures and the interpretation are unclear. Detailed comments are below.

We thank the reviewer for the interest in our work and for the useful comments that allowed us to improve the quality of the work. Specific responses are following:

1. The authors use HUVECs in the study. Given the difference of pressure between arterial and vein, the significance of using HUVECs in this study is unclear.

We thank the reviewer for this comment, that allowed us to clarify this choice. We used HUVECs as in vitro model system as in the literature the limited information available on the effects of hydrostatic pressure were mainly performed on this cell type [1-5]. We added this comment in the Results section, lines 124-127. We did it for reference and comparison purposes, but we agree with the point raised by the reviewer and in the revised version of the work we also included the analysis of the activation of YAP signaling axis and the sprouting angiogenesis in vitro in a model system of human aortic endothelial cells. As shown in the new Figure S4, the pressure stimulation of aortic endothelial cells, consistently with what observed for HUVECs, slightly modified the protein level of cadherins, relocalized YAP in the nucleus and activated its transcriptional activity. Moreover, in a 3D in vitro organotypic model system of human aortic endothelial cells pressure stimulation induced sprouting angiogenesis, that was prevented by VP treatment (Yap transcriptional activity inhibition). These results suggest that the mechanotransduction responses induced by hydrostatic pressure are independent of the cell types.

Based on these new results, these observations are now commented on in the Discussion section, lines 394-400.

2. Ki67 staining is used for evaluation of proliferation. Analysis of DNA synthesis using BrdU incorporation is helpful.

To meet the concern of the reviewer we also quantified DNA synthesis in live cells. The fluorescent detection of EdU incorporation in live cells further supported the observations obtained by staining the cells with Ki-67 nuclear cell cycle marker (Fig. 3a, b). Hydrostatic pressure-stimulated samples have a significant increase in cell proliferation compared to control. These results are reported in the results section, lines 128-137. Figure 3 has been updated, and in panel a) merged images of Ki-67+/DAPI and EdU+/DAPI are now reported.

3. Cell-cell junction integrity image (Fig. 4a) and VE-cadherin image (Fig. 4c) are not consistent. Also the difference in junctional stability among pressure is not clear. Electron microscopy would show better junctional structure.

We thank the reviewer for this comment that allows us to clarify. Images in Fig.4a have been taken at low magnification (20X) to provide an understanding of the difference in the leakage of the monolayers among the different samples and conditions. The level of definition of the cell-cell contacts in these images is not sufficient to perform any analysis on the junctions, and their levels of weakening [6] in this assay is indirectly reported by the amount of biotin leakage quantified.

On the other hand, it is possible to characterize the aspect of cell-cell junctions from immunostained images acquired at higher magnifications (e.g. [7-10]) as that present in Fig.4c and d. These high-quality junction images have been used to perform this type of analysis, which we want to highlight, is in line with previous work performed on HUVEC monolayers stimulated with similar values of hydrostatic pressure [4, 5]. We added this information in the Results section lines 169-178. Moreover, besides analyzing different features of the VE-cadherin-based junctions (Fig. 4 panel d) we also analyzed the orientation of the actin fibers with respect to the cell junctions [11]. The more perpendicular orientation of the actin fibers detected in the pressurized cells (Fig. 4e-g), together with the change in morphology of the adherens junctions demonstrated that pressure induced the organization of remodeling adhesions, which is known to indicate a reduction in junctional stabilization, and these results fit with the increased permeability we observed. All together these data demonstrate that hydrostatic pressure changes the dynamic of the adherens junctions, promoting a structural weakening that allows them to become angiogenic [6]. We realized these concepts were not

clearly explained in the text. We rephrase them in the results section to make them clearer for the readers at lines: 167-189. Electron microscopy would for sure show more in detail the structure of the junctions, but this is beyond the scope of this work.

4. Changes in N-cadherin and VE-cadherin expression are unclear in Fig. 5, both cell staining and immunoblotting.

We thank the reviewer for this comment that allows us to clarify this point, that was confused in the draft. Immunofluorescence images and quantifications in Fig.5a-b show that after 24 h of pressure stimulation a low amount of N-Cadherin was mainly localized at the cell membrane in control condition, while 24 h of 55 mmHg pressure stimulation resulted in increased expression and junctional localization of N-Cadherin. A time course analysis of N- and VE-Cadherin protein expression performed by western blot revealed that N-Cadherin was significantly increased after 24 h of 55 mmHg hydrostatic pressure stimulation and returned to basal level after 48h (Fig.5c, d). In contrast, samples under 20 mmHg slightly increased VE-Cadherin levels after 2 and 24 h, but returned to basal levels after 48 h. Conversely, under 55 mmHg pressure stimulation a modest reduction of VE-Cadherin expression was observed at 48 h. Although not all the differences in cadherin expression were statistically significant, all these variations, combined with the differences detected in the structural organization of the junctions (Fig. 4), correlate with an overall weakening of the cell-cell junctions that is also confirmed by the analysis of the monolayer barrier properties. Of note, it was previously demonstrated by us and others that N-cadherin junctional localization reduced the barrier properties of endothelial monolayers in vitro [12] and in vivo [13], as the structural modifications of the cell-cell contacts also do [7, 14]. We modified the text in the Results section lines 192-215 and discussed it at lines 361-367. We also acknowledge the comment of the reviewer, and we changed the images of the blots at 24 h with one of the other independent biological experiments we performed.

5. The authors demonstrated the YAP expression at the cell-cell junctional area. However, it is extremely faint in the staining in Fig.6 and cannot be compared between control and pressured cells.

We and others previously reported the localization of YAP at cell-cell contacts in endothelial cells in vivo and in vitro [15, 16], in specific adhesive complexes, characterized by a high degree of stability (e.g. in vivo not sprouting vessels and in vitro long confluent monolayers). In all these reports, similarly to our results, the detection of YAP junctional localization by immunostainings gives a relatively faint signal compared to the cytoplasmic and nuclear signals. As shown in Fig.6a, we detected YAP junctional localization in HUVEC cell model system, although also in this case the junctional signal is weaker than the nuclear one. We acknowledged the point raised by the reviewer about a proper comparison of this type of signal, and we produced a new supplementary Fig.S5, where we reported representative masks from which we isolated these signals to perform this type of analysis, which is detailed described in the Methods section.

As it is well established that the dynamic remodeling of the cadherin complex contributes to modulate the intracellular localization and signaling of several transcription factors among which YAP, we highlighted in the text of the Results section the more striking analysis of YAP nuclear translocation and signaling upon monolayer pressurization, and we tune down that of the reduction in junctional localization, lines 236-239.

6. In Fig.8, the authors used another assay to examine the effects of VP on endothelial sprouting. It is not clear why they switched the assay from the aortic ring assay.

We decided to perform the VP inhibition in a 3D in vitro organotypic blood vessel model of HUVECs, as we used this cell model system to identify the role of YAP signaling axis in mediating hydrostatic pressure mechanoresponses in vitro. We add this statement in the Results section, lines 295-298. For consistency, we acknowledged the advice of the reviewer, and we also performed VP treatment (YAP transcriptional activity inhibition) in 3D aortic ring assay. As now shown in Fig. 8d, we confirmed that also in aortic ring assay VP treatment was preventing the 20 mmHg samples from sprouting.

Reviewer #2 (Remarks to the Author):

This study provides a comprehensive investigation into the influence of hydrostatic pressure on sprouting angiogenesis by examining various cellular functions using in vitro and ex vivo models. The tested cellular functions include proliferation, directional migration, motility, permeability, junctional stability, EndMT, and YAP expression. The findings demonstrate that hydrostatic pressure significantly affects sprouting angiogenesis through these cellular activities. This work sheds light on complex endothelial cell physiology under physiological-

related hydrostatic pressure levels, which is significant to the angiogenesis field and beyond. It can be considered for publication after addressing the following comments.

We thank the reviewer for the interest in our work and for the useful comments that allowed us to improve the quality of the work. Specific responses are following:

1. As mentioned in this paper, other factors such as shear stress and growth factors play important roles in sprouting angiogenesis. The local mechanical environment could vary for angiogenesis such as in the wound healing process. To this end, have the co-effect of hydrostatic pressure and the other factors been considered, and would that potentially change conclusions presented in this work, for example, the conclusion that capillary hydrostatic pressure promotes sprouting while arteriole hydrostatic pressure inhibits it?

This is an important observation. Sprouting angiogenesis is regulated by biochemical and mechanical cues acting in coordination, and we agree that other factors could vary the mechanical microenvironment during angiogenesis. For example, it has been reported that for the case of wound healing angiogenesis, a crucial contribution comes from the tissue tension generated by myofibroblast contraction that pulls vessels from their original vascular beds expanding the vasculature in the growing granulation tissue [17]. In this work we studied the effect of hydrostatic pressure stimulation alone on endothelial angiogenic responses, but we need to take into consideration that in vivo endothelial cells are constantly subjected to a complex set of mechanical, chemical, and biological factors that are often highly dynamic in nature and that they integrate all these information to regulate the angiogenic functions in different types of vessels. This crosstalk between signalling induced by multiple mechanical stimuli and by bioactive molecules could have both synergistic or antagonistic effects in regulating the same cellular process that could explain why, although with less efficacy than the 20 mmHg also the 55 mmHg stimulation alone show a certain degree of sprouting capacity compared to static control. Single mechanical stimulation studies, which are now the standard way to approach mechanobiology research as reviewed by [18, 19], have the limitation that they do not reflect complex and variable physiological conditions in vivo. Alternative methods to better represent the complexity of the mechanical environment would require the development and use of devices where multiple mechanical stimuli can be applied simultaneously. This will be part of our future research direction. As we acknowledge the importance of these consideration, a statement regarding the single stimuli application was added in the Introduction (lines 57-59), and an extensive comment in the last part of the discussion section, where we highlight the limitation of this research, in lines 399-412.

2. Presented in Figures 3 -5, hydrostatic pressure (statistical) significantly improved cell motility and the impact was pressure magnitude dependent. This is also demonstrated for cadherin switching. However, a higher (55 mmHg) hydrostatic pressure does not increase paracellular leakage, which does not match the observations mentioned above. This should be noted and discussed.

We thank the reviewer for this observation. Panel b of Figure 4 in the first submitted version of the paper reported a significant increase in permeability of 20 mmHg pressurized cells compared to the control samples, and a trend in this increase for the 50 mmHg condition, which was at the limit of being statistically significant. This was due to the high dispersion of the value data and the sample size. We thus repeated the assay and we got a statistically significant increase of the 55 mmHg condition, similar to that of 20 mmHg, compared to control.

3. As discussed in the paragraph starting at line 213, no difference was identified with β -Catenin staining, but according to figure S1-a, the phospho- β -Catenin signal significantly increased under 55 mmHg after 24h. Please elaborate on the variation observed here. We thank the reviewer for this comment that allowed us to clarify the non-activation of the β -Catenin signaling pathway in cell under the various pressure stimulation we applied. In Fig. S1a, the phospho- β -Catenin signal detected after 24h of 55 mmHg stimulation looks fuzzy and dispersed in the cytoplasm compared to the other sample conditions, and parallels that of total β -Catenin, further supporting the evidence that there is a weakening in the junctional organization under pressure. To answer the reviewer's concern, we performed an additional western blot analysis of the phospho- and total levels of β -Catenin. As now shown in the new panels b and c of Fig. S1 no differences were observed.

Beside clarifying and integrating the immunofluorescence data, these new results are in line with same level of activation of the GSK3 β (Fig. S1d, e), which regulates β -catenin signaling

activity, and the same transcriptional level of expression of Axin2 (Fig. S1f), a β -catenin target gene, that acts in a negative feedback loop to limit and fine-tune β -catenin signaling. Overall, the new data presented show that the values of static hydrostatic pressure stimulation we applied did not impact on β -catenin signaling.

4. In this study, an ex vivo 3D aortic ring assay, a 2D in vitro HUVECs model, and a 3D in vitro organotypic blood vessel model were used to test sprouting angiogenesis and different cellular activities. Have the variations between the models been assessed and shown to support a consistent line of research? For example, does hydrostatic pressure have the same impact on human v.s. mouse UVEC? Has any other assay been considered, for example, HDMEC sprouting, tube formation, or any in vivo assays?

Blood pressure value is comparable between mice and humans [20-23]. Previous reports showed that hydrostatic pressure has the same impact on endothelial cells of human and mouse models [24, 25]. Except for the aortic ring assay, which was performed using murine samples, all the other in vitro 3D and 2D assays were performed using HUVECs as model system. The reason for this choice was that in the literature the limited information available on the effects of hydrostatic pressure were mainly performed on this cell type [1-5] and we did it for reference and comparison purposes. In our work, consistent phenotypic results were obtained in 3D ex vivo mouse aortic ring assay and 3D in vitro organotypic HUVEC model. In both types of assays, we detected an increase in sprouting angiogenesis under pressure stimulation, being the 20 mmHg stimulation the most effective. To further improve the consistency between the models, in the revised version of the paper we included the VP treatment (YAP transcriptional activity inhibition), previously performed only in the 3D in vitro organotypic HUVEC assay, also in the ex vivo aortic ring assay (Fig. 8d, e). Results show that YAP inhibition prevented hydrostatic pressure-induced sprouting angiogenesis also in the murine model.

Last, in the revised version of the work, we also included the analysis of the activation of YAP signaling axis and the sprouting angiogenesis in a 3D in vitro organotypic model system of human aortic endothelial cells (Fig. S4h, i), that supported the consistency of our line of research.

Reviewer #3 (Remarks to the Author):

In this work Al-Nuaimi and co-authors study the impact of hydrostatic pressure on endothelial cells. They utilize mouse aortic endothelial cells, in 3D aortic ring assays, and human umbilical vein endothelial cells (HUVECs) to study angiogenic sprouting, proliferation, migration and junctional barrier integrity under hydrostatic pressure. The experiments are conducted under 0 mmHg (control), 20 mmHg (mean capillary pressure) and 55 mmHg (peripheral arteriole pressure). The authors further show that N-cadherin is transiently increased in HUVECs under induced hydrostatic pressure only after 24h and that YAP1 is localized to the nucleus. Accordingly, some YAP target genes are regulated under selected hydrostatic pressure conditions and Verteporfin treatment of HUVECs in a 3D organotypic model was able to restrict 20 mmHg-induced sprouting.

While the impact of hydrostatic pressure on endothelial cell functions, such as proliferation and junctional barrier maintenance has already been studied, the authors observe some additional new findings. The experiments are technically convincing, and the manuscript is well written. However, this reviewer feels that the obtained data is insufficiently discussed and are to some extent overinterpreted. Selected additional experiments would strengthen the manuscript.

We thank the reviewer for the interest in our work and for the useful comments that allowed us to improve the quality of the work. Specific responses are following:

Major points:

1. While authors use arterial and venous endothelial cells, they do not analyze or discuss endothelial cell type differences throughout the manuscript. Aortic rings are dissected from mouse aortas, where the endothelial cells certainly experience higher hydrostatic pressure as the pressure values, they are subsequently subjected to in the aortic ring assay. Why do 55 mmHg show higher sprouting capacity than 0 mmHg controls? Should high pressures not prevent sprouting in aortic endothelial cells? In that context, how does venous capillary pressure massively increase sprouting in comparison to 0 mmHg controls? The authors should perform key experiments, such as YAP nuclear localization, YAP target gene expression and 3D organotypic model experiments with human aortic endothelial cells to evaluate potential endothelial cell type differences.

We agree with the reviewer and in the revised version of the paper endothelial cell type differences have been studied and discussed.

Except for the aortic ring assay, which was performed using murine samples of arterial origin, all the other in vitro 3D and 2D assays were performed using HUVECs as model system. The reason for this choice was that in the literature the limited information available on the effects

of hydrostatic pressure were mainly performed on this cell type [1-5] and we did it for reference and comparison purposes. In our work, consistent phenotypic results were obtained in 3D ex vivo mouse aortic ring assay and 3D in vitro organotypic HUVEC model. In both types of assays, we detected an increase in sprouting angiogenesis under pressure stimulation, being the 20 mmHg stimulation the most effective. To further improve the consistency between the models, in the revised version of the paper we included the VP treatment (YAP inhibition), previously performed only in the 3D in vitro assay, also in the ex vivo aortic ring assay (Fig. 8d, e). Results show that YAP inhibition prevented hydrostatic pressure-induced sprouting angiogenesis also in the arterial cell model.

Last, as suggested by the reviewer, we also included the analysis of the activation of YAP signalling axis and the sprouting angiogenesis in a 3D in vitro model system of human aortic endothelial cells (Fig. S4h, i).

The new set of experiments performed on this different cell type showed that the hydrostatic pressure stimulation alone was able to increase YAP nuclear localization and signalling, in a way like that observed in endothelial cells of venous origin. These results suggest that the considered values of hydrostatic pressure stimulation exert similar effects on endothelial cells independent of the cell types, and this could lie in the high degree of plasticity of these cells. Endothelial cells in vivo possess great phenotypic variability depending on the vascular bed and organs where they are located, and mechanical signals are known contributors of the heterogeneity of endothelial phenotypes [26]. On the other hand, a lot of literature reports about the role of mechanical forces as modulators of the endothelial plasticity, the remarkable ability to change phenotypic characteristics and functions to adapt to various stimuli, at molecular, cellular and tissue levels. In adult tissues, endothelial plasticity is considered the driving force of homeostasis and remodelling in physiology, and of pathological angiogenesis. It has been previously reported that although the acquisition of arterial and venous identity is established during development by genetically programmed pathways, changes of hemodynamic conditions and substrate mechanics are able to modify and even switch phenotypes and functions of primary endothelial cells of different origin. Changes in the pattern of flow convert veins into arteries and vice versa in vivo [27-29], and in vitro [30, 31]. Saphenous vein grafts switch their cell identity when placed on arteries, downregulating venous markers, and expressing arterial ones, while they keep their venous signature when placed on veins [27]. During vascular remodelling in zebrafish embryos, arterial cells contribute to venous vessels, and venous cells reprogram into arterial cells [32, 33]. Arterial-venous endothelial identity can be reprogrammed by topography, as primary endothelial cells of venous and arterial origins show a venous phenotype when cultured on grooves [34]. Arterial endothelial cells switch to a venous phenotype when culture on soft substrates [35]. It has also been reported that stretch stimulation equally modulates the mechano-responses of different endothelial cell types, as both arterial- and venous-derived primary endothelial cells increased their proliferation under cyclic stretch [36-39]. Overall, our results suggest the mechanotransduction-induced responses by hydrostatic pressure stimulation trigger similar effects on cells of different origins, supporting the concept that that endothelial morphology and function are mostly determined by the biomechanical properties of the environment rather than genetically determined differences. Last, in this work we studied the effect of hydrostatic pressure stimulation alone on endothelial angiogenic responses, but we need to take into consideration that in vivo endothelial cells are constantly subjected to a complex set of mechanical, chemical, and biological factors that are often highly dynamic in nature and that they integrate all these information to regulate the angiogenic functions in different types of vessels. This crosstalk between signalling induced by multiple mechanical stimuli and by bioactive molecules could have both synergistic or antagonistic effects in regulating the same cellular process that could explain why, although with less efficacy than the 20 mmHg also the 55 mmHg stimulation alone show a certain degree of sprouting capacity compared to static control. Single mechanical stimulation studies, which are now the standard way to approach mechanobiology research (as reviewed by [18, 19]), have the limitation that they do not reflect complex and variable physiological conditions in vivo. Alternative methods to better represent the complexity of the mechanical environment would require the development and use of devices where multiple mechanical stimuli can be applied simultaneously. This will be part of our future research direction. As we acknowledge the importance of these considerations, we extensively commented on them in the discussion, lines 389-412.

2. Why aortic rings and 3D organotypic models do provide endothelial cells with a compliant/soft or more physiological stiffness environment, presumably standard endothelial cell culture dishes do not enable endothelial cells to accurately react to hydrostatic pressure. The authors should discuss this caveat as YAP/TAZ are regulated in endothelial cells via the stiffness microenvironment, which is obviously different between the here compared

experimental approaches.

We thank the reviewer for this useful observation. The localization and signaling of YAP in endothelial cells are regulated by various mechanisms. These include cell-cell contact inhibition [15, 16, 40], mechanical forces such as shear stress [41] and substrate stiffness [42] that modulate signaling pathways such as the Hippo and the VEGF/VEGFR pathways and the actin cytoskeleton dynamics. In the revised version of this paper, we provide additional results showing that hydrostatic pressure stimulation was also triggering nuclear relocalization in endothelial monolayers cultured on soft and more compliant substrates (collagen hydrogel of ~ 3-4 kPa stiffness, [43]). These results are now included in the new Fig. S2, and discussed in the Results section in lines 240-248.

3. In line 82 the authors state that transient increased N-cadherin promotes the weakening of adherens junctions. This statement is not confirmed in an experimental setup and simple is assumed based on correlation. Furthermore, N-Cadherin, which in endothelial monolayers is typically localized at the cell membrane, was upregulated and strongly re-localized to cell-cell contacts (Line 311-312). This statement cannot be seen from the images provided in Figure 5a. The Western blot in Figure 5c solely shows N-cadherin protein upregulation after 24h in the 55 mmHg-condition, which does not correlate with highest monolayer permeability under 20mmHg pressure in Figure 4a. Why does VE-cadherin not change on the protein level at all? We thank the reviewer for this comment that allows us to clarify this point. Immunofluorescence images and quantifications in Fig.5a-b show that after 24 h of pressure stimulation a low amount of N-Cadherin was mainly localized at the cell membrane in control condition, while 24 h of 55 mmHg pressure stimulation resulted in increased expression and junctional localization of N-Cadherin. A time course analysis of N- and VE-Cadherin expression performed by western blot revealed that N-Cadherin was significantly increased after 24 h of 55 mmHg hydrostatic pressure stimulation and returned to basal level after 48h (Fig.5c, d). In contrast, samples under 20 mmHg slightly increased VE-Cadherin levels after 2 and 24 h, but returned to basal levels after 48 h. Conversely, under 55 mmHg pressure stimulation a modest reduction of VE-Cadherin expression was observed at 48 h. Although not all the differences in cadherin localization and expression were statistically significant, all these variations, combined with the differences detected in the structural organization of the junctions (Fig.4), correlate with an overall weakening of the cell-cell junctions we further confirmed by analysing monolayer barrier properties. We agree with the reviewer that this is a correlation, but it was already previously demonstrated by us and others that N-cadherin junctional localization reduced the barrier properties of endothelial monolayers in vitro [12] and in vivo [13], as the structural modifications of the cell-cell contacts also do [7, 14]. Finally, the fact that we did not detect a strong downregulation of VE-Cadherin is in line with the partial EndMT phenotype observed, as in this specific phenomenon, endothelial cells acquire migratory characteristics while remaining connected to neighbouring cells (Fang, 2021). According to the reviewer comments, the Results and Discussion sections have been modified.

4. The transient nature of the in vitro phenotypes observed in this work should be addressed in more detail and related to potential in vivo situations. How long have the cells been cultured before they are subjected to pressure? What was the status of confluency?

All the experiments were performed after the establishment of mature monolayers, obtained by culturing confluent cells for 72 h [15]. Before seeding them in the different experimental configurations, frozen vials were thawed, cells were splitted one time, and cultured/expanded in static conditions for about 1 week. This information is now clearly reported in the cell culture method section of the revised paper, lines 423-436.

Cell adaptation to stimuli can be both transient or long-lasting, depending on the duration and magnitude of the stimulus. It was previously reported that in the vascular system, mechanical signals can rapidly activate intracellular signalling pathways, the duration of which are typically short-lived, as the transient nature of these signals allows cells to accommodate the new mechanical state and adapt their behaviour accordingly to prevent endothelial damage. The reversibility of these adaptations may vary depending on the specific mechanical stimulus applied (physiological vs pathological). In our work, we observed that upon hydrostatic pressure stimulation, our cells transiently activate YAP signalling axis, acquire a partial EndMT phenotype characterised by the presence of both endothelial and mesenchymal features, and activate sprouting angiogenesis programs. It was previously reported by many different groups that one of main characteristics of the partial EndMT event is that it is transient [44, 45], as this type of activation in healthy and diseased angiogenesis, prevents excessive mesenchymal transition and sustains organized and controlled new vessel growth [30, 46]. We acknowledge the importance of commenting this aspect that has now been included in the discussion section in lines 368-389.

Minor points:

1. In the supplemental data S2c the authors show that YAP is already increased after 48h in

the bioreactor pressure control versus conventional cell culture pressure. Could the authors show what is the YAP expression after 24h in this experiment, as this is the usual time point studied in the main manuscript.

We apologize this time point was missing, the experiments have been repeated including the 24 h time point and as shown now in the new Fig.S3, no significant differences in YAP expression were detected among all the time points analyzed.

2. In Figure 6e ANKRD1 is not up-regulated after 24h with 20 mmHg but it is in Figure 8a. Can the authors explain the difference?

ANKRD1 is not upregulated in Fig. 6e as well as in Fig. 8a (p values is not < 0.05). What is statistically significant in Fig.8a is the ANKRD1 downregulation upon VP treatment (YAP transcriptional activity inhibition). We agree with the reviewers that there are small differences in the 20 mmHg at 24h between Fig.6e and 8a, but this could be likely due to the facts that all control samples in the experiment of VP treatment have been treated with DMSO as vehicle. We highlighted that controls were treated with DMSO in the Results section, lines 292-295.

3. Figure 4e control and Figure 5a control show the same field of view. The authors should indicate that the information has been obtained from the same experiments.

The reviewer is right. As all our samples have been stained with phalloidin and junctions, to perform the analysis of Fig. 4e we used all the images from different sets of independent experiments. We added this comment in the methods sections (lines 631-633), but not to create confusion in the readers we changed the image in Fig. 5a.

4. Statements, such as "for the first time" (Line 35), should not be used. The statement has been removed.

References

1. Yoshino, D., K. Funamoto, K. Sato, Kenry, M. Sato, and C.T. Lim, Hydrostatic pressure promotes endothelial tube formation through aquaporin 1 and Ras-ERK signaling. *Commun Biol*, 2020. 3(1): p. 152.
2. Shen, H.X., J.Z. Liu, X.Q. Yan, H.N. Yang, S.Q. Hu, X.L. Yan, T. Xu, A.J. El Haj, Y. Yang, and L.X. Lu, Hydrostatic pressure stimulates the osteogenesis and angiogenesis of MSCs/HUVECs co-culture on porous PLGA scaffolds. *Colloids Surf B Biointerfaces*, 2022. 213: p. 112419.
3. Schwartz, E.A., R. Bizios, M.S. Medow, and M.E. Gerritsen, Exposure of human vascular endothelial cells to sustained hydrostatic pressure stimulates proliferation - Involvement of the alpha(v) integrins. *Circulation Research*, 1999. 84(3): p. 315-322.
4. Ohashi, T., K. Segawa, N. Sakamoto, and M. Sato, Effect of Hydrostatic Pressure on the Morphology and Expression of VE-Cadherin in HUVEC. *Transactions of Japanese Society for Medical and Biological Engineering*, 2006. 44(3): p. 454-459.
5. Ohashi, T., Y. Sugaya, N. Sakamoto, and M. Sato, Hydrostatic pressure influences morphology and expression of VE-cadherin of vascular endothelial cells. *J Biomech*, 2007. 40(11): p. 2399-405.
6. Lampugnani, M.G. and E. Dejana, Adherens junctions in endothelial cells regulate vessel maintenance and angiogenesis. *Thromb Res*, 2007. 120 Suppl 2: p. S1-6.
7. Huvneers, S., J. Oldenburg, E. Spanjaard, G. van der Krogt, I. Grigoriev, A. Akhmanova, H. Rehmann, and J. de Rooij, Vinculin associates with endothelial VE-cadherin junctions to control force-dependent remodeling. *J Cell Biol*, 2012. 196(5): p. 641-52.
8. Oldenburg, J. and J. de Rooij, Mechanical control of the endothelial barrier. *Cell Tissue Res*, 2014. 355(3): p. 545-55.
9. Maeso-Alonso, L., H. Alonso-Olivares, N. Martnez-Garca, L. Lpez-Ferreras, J. Villoch-Fernndez, L. Puente-Santamara, N. Colas-Algora, A. Fernndez-Corona, M.E. Lorenzo-Marcos, B. Jimnez, L. Holmgren, M. Wilhelm, J. Millan, L. del Peso, L. Claesson-Welsh, M.M. Marques, and M.C. Marin, p73 is required for vessel integrity controlling endothelial junctional dynamics through Angiotin. *Cellular and Molecular Life Sciences*, 2022. 79(10).
10. Jin, Y., Y. Ding, M. Richards, M. Kaakinen, W. Giese, E. Baumann, A. Szyborska, A. Rosa, S. Nordling, L. Schimmel, E.B. Akmeric, A. Pena, E. Nwadozi, M. Jamalpour, K. Holstein, M. Sainz-Jaspeado, M.O. Bernabeu, M. Welsh, E. Gordon, C.A. Franco, D. Vestweber, L. Eklund, H. Gerhardt, and L. Claesson-Welsh, Tyrosine-protein kinase Yes controls endothelial junctional plasticity and barrier integrity by regulating VE-cadherin phosphorylation and endocytosis. *Nat Cardiovasc Res*, 2022. 1(12): p. 1156-1173.
11. Dorland, Y.L. and S. Huvneers, Cell-cell junctional mechanotransduction in

- endothelial remodeling. *Cell Mol Life Sci*, 2017. 74(2): p. 279-292.
12. Taddei, A., C. Giampietro, A. Conti, F. Orsenigo, F. Breviario, V. Pirazzoli, M. Potente, C. Daly, S. Dimmeler, and E. Dejana, Endothelial adherens junctions control tight junctions by VE-cadherin-mediated upregulation of claudin-5. *Nat Cell Biol*, 2008. 10(8): p. 923-34.
 13. Luo, Y. and G.L. Radice, N-cadherin acts upstream of VE-cadherin in controlling vascular morphogenesis. *J Cell Biol*, 2005. 169(1): p. 29-34.
 14. Rabiet, M.J., J.L. Plantier, Y. Rival, Y. Genoux, M.G. Lampugnani, and E. Dejana, Thrombin-induced increase in endothelial permeability is associated with changes in cell-to-cell junction organization. *Arterioscler Thromb Vasc Biol*, 1996. 16(3): p. 488-96.
 15. Giampietro, C., A. Disanza, L. Bravi, M. Barrios-Rodiles, M. Corada, E. Frittoli, C. Savorani, M.G. Lampugnani, B. Boggetti, C. Niessen, J.L. Wrana, G. Scita, and E. Dejana, The actin-binding protein EPS8 binds VE-cadherin and modulates YAP localization and signaling. *J Cell Biol*, 2015. 211(6): p. 1177-92.
 16. Neto, F., A. Klaus-Bergmann, Y.T. Ong, S. Alt, A.C. Vion, A. Szymborska, J.R. Carvalho, I. Hollfinger, E. Bartels-Klein, C.A. Franco, M. Potente, and H. Gerhardt, YAP and TAZ regulate adherens junction dynamics and endothelial cell distribution during vascular development. *Elife*, 2018. 7.
 17. Kilarski, W.W., B. Samolov, L. Petersson, A. Kvanta, and P. Gerwins, Biomechanical regulation of blood vessel growth during tissue vascularization. *Nat Med*, 2009. 15(6): p. 657-64.
 18. Dessalles, C.A., C. Leclech, A. Castagnino, and A.I. Barakat, Integration of substrate- and flow-derived stresses in endothelial cell mechanobiology. *Commun Biol*, 2021. 4(1): p. 764.
 19. Gordon, E., L. Schimmel, and M. Frye, The Importance of Mechanical Forces for in vitro Endothelial Cell Biology. *Front Physiol*, 2020. 11: p. 684.
 20. Poulsen, C.B., T. Wang, K. Assersen, N.K. Iversen, and M. Damkjaer, Does mean arterial blood pressure scale with body mass in mammals? Effects of measurement of blood pressure. *Acta Physiol (Oxf)*, 2018. 222(4): p. e13010.
 21. Sandal, P.H., M. Damgaard, and N.H. Secher, Comments on the Review 'Does mean arterial blood pressure scale with body mass in mammals? Effect of measurement of blood pressure' *Acta Physiol (Oxf)*. *Acta Physiol (Oxf)*, 2020. 228(1): p. e13407.
 22. White, C.R. and R.S. Seymour, The role of gravity in the evolution of mammalian blood pressure. *Evolution*, 2014. 68(3): p. 901-8.
 23. Dawson, T.H., Allometric Relations and Scaling Laws for the Cardiovascular System of Mammals. *Systems*, 2014. 2(2): p. 168-185.
 24. Friedrich, E.E., Z. Hong, S. Xiong, M. Zhong, A. Di, J. Rehman, Y.A. Komarova, and A.B. Malik, Endothelial cell Piezo1 mediates pressure-induced lung vascular hyperpermeability via disruption of adherens junctions. *Proc Natl Acad Sci U S A*, 2019. 116(26): p. 12980-12985.
 25. Mammoto, T., T. Hunyenyiwa, P. Kyi, K. Hendee, K. Matus, S. Rao, S.H. Lee, D.M. Tabima, N.C. Chesler, and A. Mammoto, Hydrostatic Pressure Controls Angiogenesis Through Endothelial YAP1 During Lung Regeneration. *Front Bioeng Biotechnol*, 2022. 10: p. 823642.
 26. Urner, S., M. Kelly-Goss, S.M. Peirce, and E. Lammert, Mechanotransduction in Blood and Lymphatic Vascular Development and Disease. *Adv Pharmacol*, 2018. 81: p. 155-208.
 27. Bai, H., Z. Wang, M. Li, P. Sun, S. Wei, Z. Wang, Y. Xing, and A. Dardik, Adult Human Vein Grafts Retain Plasticity of Vessel Identity. *Ann Vasc Surg*, 2020. 68: p. 468-475.
 28. le Noble, F., D. Moyon, L. Pardanaud, L. Yuan, V. Djonov, R. Matthijsen, C. Breant, V. Fleury, and A. Eichmann, Flow regulates arterial-venous differentiation in the chick embryo yolk sac. *Development*, 2004. 131(2): p. 361-75.
 29. Moyon, D., L. Pardanaud, L. Yuan, C. Breant, and A. Eichmann, Plasticity of endothelial cells during arterial-venous differentiation in the avian embryo. *Development*, 2001. 128(17): p. 3359-70.
 30. Fang, J.S., B.G. Coon, N. Gillis, Z. Chen, J. Qiu, T.W. Chittenden, J.M. Burt, M.A. Schwartz, and K.K. Hirschi, Shear-induced Notch-Cx37-p27 axis arrests endothelial cell cycle to enable arterial specification. *Nat Commun*, 2017. 8(1): p. 2149.
 31. Mack, J.J., T.S. Mosqueiro, B.J. Archer, W.M. Jones, H. Sunshine, G.C. Faas, A. Briot, R.L. Aragn, T. Su, M.C. Romay, A.I. McDonald, C.H. Kuo, C.O. Lizama, T.F. Lane, A.C. Zovein, Y. Fang, E.J. Tarling, T.Q.D. Vallim, M. Navab, A.M. Fogelman, L.S. Bouchard, and M.L. Iruela-Arispe, NOTCH1 is a mechanosensor in adult arteries. *Nature Communications*, 2017. 8.
 32. Quillien, A., J.C. Moore, M. Shin, A.F. Siekmann, T. Smith, L. Pan, C.B. Moens, M.J. Parsons, and N.D. Lawson, Distinct Notch signaling outputs pattern the developing arterial system. *Development*, 2014. 141(7): p. 1544-52.
 33. Red-Horse, K., H. Ueno, I.L. Weissman, and M.A. Krasnow, Coronary arteries form by developmental reprogramming of venous cells. *Nature*, 2010. 464(7288): p. 549-53.

34. Arora, S., S. Lin, C. Cheung, E.K.F. Yim, and Y.C. Toh, Topography elicits distinct phenotypes and functions in human primary and stem cell derived endothelial cells. *Biomaterials*, 2020. 234: p. 119747.
35. Fioretta, E.S., J.O. Fledderus, F.P. Baaijens, and C.V. Bouten, Influence of substrate stiffness on circulating progenitor cell fate. *J Biomech*, 2012. 45(5): p. 736-44.
36. Li, W. and B.E. Sumpio, Strain-induced vascular endothelial cell proliferation requires PI3K-dependent mTOR-4E-BP1 signal pathway. *Am J Physiol Heart Circ Physiol*, 2005. 288(4): p. H1591-7.
37. Nishimura, K., W. Li, Y. Hoshino, T. Kadohama, H. Asada, S. Ohgi, and B.E. Sumpio, Role of AKT in cyclic strain-induced endothelial cell proliferation and survival. *Am J Physiol Cell Physiol*, 2006. 290(3): p. C812-21.
38. Matsumoto, T., Y.C. Yung, C. Fischbach, H.J. Kong, R. Nakaoka, and D.J. Mooney, Mechanical strain regulates endothelial cell patterning in vitro. *Tissue Eng*, 2007. 13(1): p. 207-17.
39. Kou, B., J. Zhang, and D.R. Singer, Effects of cyclic strain on endothelial cell apoptosis and tubulogenesis are dependent on ROS production via NAD(P)H subunit p22phox. *Microvasc Res*, 2009. 77(2): p. 125-33.
40. Choi, H.J., H. Zhang, H. Park, K.S. Choi, H.W. Lee, V. Agrawal, Y.M. Kim, and Y.G. Kwon, Yes-associated protein regulates endothelial cell contact-mediated expression of angiopoietin-2. *Nat Commun*, 2015. 6: p. 6943.
41. Zhang, Q., Y. Cao, Y. Liu, W. Huang, J. Ren, P. Wang, C. Song, K. Fan, L. Ba, L. Wang, and H. Sun, Shear stress inhibits cardiac microvascular endothelial cells apoptosis to protect against myocardial ischemia reperfusion injury via YAP/miR-206/PDCD4 signaling pathway. *Biochem Pharmacol*, 2021. 186: p. 114466.
42. Matsuo, E., T. Okamoto, A. Ito, E. Kawamoto, K. Asanuma, K. Wada, M. Shimaoka, M. Takao, and A. Shimamoto, Substrate stiffness modulates endothelial cell function via the YAP-Dll4-Notch1 pathway. *Exp Cell Res*, 2021. 408(1): p. 112835.
43. Wahlsten, A., A. Stracuzzi, I. Luchtefeld, G. Restivo, N. Lindenblatt, C. Giampietro, A.E. Ehret, and E. Mazza, Multiscale mechanical analysis of the elastic modulus of skin. *Acta Biomater*, 2023. 170: p. 155-168.
44. Fang, J.S., N.W. Hultgren, and C.C.W. Hughes, Regulation of Partial and Reversible Endothelial-to-Mesenchymal Transition in Angiogenesis. *Front Cell Dev Biol*, 2021. 9: p. 702021.
45. Tombor, L.S., D. John, S.F. Glaser, G. Luxan, E. Forte, M. Furtado, N. Rosenthal, N. Baumgarten, M.H. Schulz, J. Wittig, E.M. Rogg, Y. Manavski, A. Fischer, M. Muhly-Reinholz, K. Klee, M. Looso, C. Selignow, T. Acker, S.I. Bibli, I. Fleming, R. Patrick, R.P. Harvey, W.T. Abplanalp, and S. Dimmeler, Single cell sequencing reveals endothelial plasticity with transient mesenchymal activation after myocardial infarction. *Nat Commun*, 2021. 12(1): p. 681.
46. Hultgren, N.W., J.S. Fang, M.E. Ziegler, R.N. Ramirez, D.T.T. Phan, M.M.S. Hatch, K.M. Welch-Reardon, A.E. Paniagua, L.S. Kim, N.N. Shon, D.S. Williams, A. Mortazavi, and C.C.W. Hughes, Slug regulates the Dll4-Notch-VEGFR2 axis to control endothelial cell activation and angiogenesis. *Nat Commun*, 2020. 11(1): p. 5400.

Version 1:

Reviewer comments:

Reviewer #1

(Remarks to the Author)

The authors responded to my comments appropriately.

Reviewer #2

(Remarks to the Author)

Thank you for addressing my comments. I do not have any further questions.

Reviewer #3

(Remarks to the Author)

The authors have addressed most of my concerns.

Two points should be considered:

1) In vitro findings using HUVECs and arterial endothelial cells cultured on plastic dishes should not be used to make a bold statement "These results support the concept that endothelial functions are mostly determined by the biomechanical properties of the environment". This statement should be toned down or removed.

2) The experiments performed on compliant hydrogels (cell type not indicated in the S2 figure legend) already demonstrate that by changing the substrate stiffness the pressure-induced phenotype (YAP relocalization, junctional dispersion/orientation) is less severe in a softer environment. This should be included in the discussion and future directions.

Author Rebuttal letter:

COMMSBIO-23-4298: Rebuttal letter

Reviewers' comments:

Reviewer #1 (Remarks to the Author):

The authors responded to my comments appropriately.

We thank the reviewer one more time for the provided inputs that have helped us to improve the quality and clarity of our work.

Reviewer #2 (Remarks to the Author):

Thank you for addressing my comments. I do not have any further questions.

We thank the reviewer one more time for the provided inputs that have helped us to improve the quality and clarity of our work.

Reviewer #3 (Remarks to the Author):

The authors have addressed most of my concerns.

Two points should be considered:

1) In vitro findings using HUVECs and arterial endothelial cells cultured on plastic dishes should not be used to make a bold statement "These results support the concept that endothelial functions are mostly determined by the biomechanical properties of the environment". This statement should be toned down or removed.

We thank the reviewer for this comment, and we toned down this statement in the discussion (pag. 14-15)

2) The experiments performed on compliant hydrogels (cell type not indicated in the S2 figure legend) already demonstrate that by changing the substrate stiffness the pressure-induced phenotype (YAP relocalization, junctional dispersion/orientation) is less severe in a softer environment. This should be included in the discussion and future directions.

We apologize, the cell type was indicated only in the 2D cell culture on soft collagen gels paragraph of the Methods. We now added it also in the figure legend of Figure S2.

We acknowledge the suggestion of the reviewer and the less severe phenotype on softer environment has been now highlighted and commented in the Result section (pag. 9) and in the Discussion (pag.14).
